# Posterior Collapse of a Linear Latent Variable Model

**Zihao Wang**[*]
Department of CSE
HKUST

**Liu Ziyin**[*]
Department of Physics
The University of Tokyo

## Abstract

This work identifies the existence and cause of a type of posterior collapse that frequently occurs in the Bayesian deep learning practice. For a general linear latent variable model that includes linear variational autoencoders as a special case, we precisely identify the nature of posterior collapse to be the competition between the likelihood and the regularization of the mean due to the prior. Our result suggests that posterior collapse may be related to neural collapse and dimensional collapse and could be a subclass of a general problem of learning for deeper architectures.

## 1 Introduction

Bayesian approaches to deep learning have attracted much attention because they allow for a more principled treatment of inference and uncertainty estimation (Mackay, 1992; Neal, 2012; Wang and Yeung, 2020; Jiang and Ahn, 2020; Zhao et al., 2021; Liu, 2021). One long-standing and unresolved problem for the Bayesian deep learning practice is the problem of posterior collapse, where the posterior distribution of the learned latent variables partially or completely collapses with the prior (Bowman et al., 2015; Huang et al., 2018; Lucas et al., 2019; Razavi et al., 2019; Kingma et al., 2016; Wang et al., 2021). Up to now, the study of the nature of the cause of the posterior collapse problem has been limited. There are two main challenges that prevent our understanding of the problem: (1) posterior collapses mainly occur in deep learning, and the landscape of deep neural networks is hard to understand in general; (2) the use of approximate loss functions such as the evidence lower bound (ELBO) complicates the problem.

Consider a problem where one wants to model the data distribution $p(x, y)$ through a latent variable $z$. The evidence lower bound (ELBO) loss function reads

$$\underbrace{\mathbb{E}_{x,y}[-\mathbb{E}_{q(z|x)}\log(p(y|z))]}_{\ell_{rec}} + \underbrace{\mathbb{E}_x[D_{KL}(q(z|x)\|p(z))]}_{\ell_{KL}}, \tag{1}$$

where $q$ is the approximate distribution, we rely on to approximate the true distribution $p$. This loss is more general than the standard ELBO for variational autoencoders (VAE) (Kingma and Welling, 2013). Meanwhile, it can be seen as the simplest type of loss for a conditional VAE (Sohn et al., 2015), where one aims to model a conditional distribution $p(y|x)$. The distribution $p(z)$ is the prior distribution of the latent variable $z$ and is often a low-complexity distribution such as a zero-mean unit-variance Gaussian. This loss function thus has a clean interpretation as the sum of a prediction accuracy term (the first term $\ell_{rec}$) that encourages better prediction accuracy and a complexity term (the second term $\ell_{KL}$) that encourages a simpler solution. Learning under this loss function proceeds by balancing the prediction error and the model simplicity. Moreover, learning under this loss function has also been used as one of the primary theoretical models in neuroscience (Friston, 2009), and its understanding may also help advance theoretical neuroscience. This work provides an in-depth study of the posterior collapse problem of Eq. (1), when the decoder $q(y|z)$ and encoder $q(z|x)$ are each parametrized by a linear model.

36th Conference on Neural Information Processing Systems (NeurIPS 2022).

---

[*]Equal contribution.

Specifically, our contributions include:

- we find the global minima of a general linear latent variable model that includes the linear VAE as a special case under the Objective (1);
- we find the precise condition when posterior collapse occurs, where the global minimum is the origin;
- we pinpoint the cause of the posterior collapse to be the excessively strong regularization effect on the *mean* of the latent variables due to the prior.

To the best of our knowledge, our work is the first to pinpoint the cause of the posterior collapse problem. This work is organized as follows. The next section discusses the previous literature. Section 3 describes the theoretical problem setting. Section 4 presents our main technical results and analyzes them in detail. Section 5 presents numerical examples. The last section concludes this work and points to the remaining open problems. The Appendix B investigates the effect of the bias term, Appendix C details the effect of a data-dependent encoder variance, and Appendix D treats the case of a learnable decoder variance.

## 2 Related Works

*Approximate Bayesian Deep Learning*. Bayesian deep learning in general and VAE training, in particular, rely heavily on approximate methods such as the ELBO objective because the exact probabilities are intractable. The connection of approximate Bayesian learning and probabilistic PCA (pPCA) has been extensively studied (Nakajima and Sugiyama, 2010; Nakajima et al., 2013, 2015; Lucas et al., 2019).

*Causes of Posterior Collapse*. Earlier touches on the problem tend to attribute the cause of posterior collapse to the use of approximate methods, namely, to the use of the ELBO (Bowman et al., 2015; Huang et al., 2018; Razavi et al., 2019). Another line of work attributes the cause to the high capacity of modern neural networks (Alemi et al., 2018; Ziyin et al., 2022c). However, Lucas et al. (2019) showed that for a simplified linear model, the ELBO is not the cause of posterior collapse because the posterior collapse exists even in the exact posterior. It also implies that the posterior collapse is not due to the high capacity of the models because linear models have a limited capacity. Lucas et al. (2019) then suggested that making the decoder variance learnable can fix the collapse problem and that an unlearnable decoder variance is the cause of the posterior collapse. However, our results show that this is not the case: for both learnable and unlearnable decoder variance, there exist situations where a collapse happens or does not happen, which implies that the learnability of the decoder variance does not have a causal relation with posterior collapse, nor is it sufficient to fix the problem (Section D). In terms of the problem setting, ours is also more general than Lucas et al. (2019) because our result (1) applies to general latent variable models (one example being the conditional VAE (Sohn et al., 2015)) and (2) considers the case of $\beta$-VAE with a general $\beta$ when the decoder variance is learned. An important implication of our work is that posterior collapses can be a ubiquitous problem for deep-learning-based latent-variable models (not just unique to autoencoding models) and that they share a common cause. Meanwhile, (Lücke et al., 2020) shows that posterior collapse can happen due to the tradeoff between the decoding performance and the decoding entropy. (Shekhovtsov et al., 2022) demonstrated the relationship between model consistency and posterior collapse and suggested that a proper choice of data processing or architecture may alleviate collapse.

*Linear Networks*. Deep linear nets have been extensively used to understand the landscape of non-linear networks. For example, linear regressors are shown to be relevant for understanding the generalization behavior of modern overparametrized networks (Hastie et al., 2019). Saxe et al. (2013) used a two-layer linear network to understand the dynamics of learning nonlinear networks. The linear nets are the same as a linear regression model in terms of expressivity. However, the loss landscape is highly complicated due to depth. (Kawaguchi, 2016; Hardt and Ma, 2016; Laurent and Brecht, 2018; Ziyin et al., 2022a). Our work essentially studies the loss landscape of linear networks. While each encoder and decoder we use consists of a single linear layer, they effectively constitute a two-layer linear network when trained together.

## 3 Problem Setting

We consider a general linear latent variable model with input space $x \in \mathbb{R}^{D_0}$, latent space $z \in \mathbb{R}^{d_1}$, and target space $y \in \mathbb{R}^{d_2}$. In general, $y = f(x)$ is an arbitrary function of $x$. When the target $y$ is identical to the input $x$, it reduces to the standard VAE. The VAE formalism assumes that there

is an intermediate "latent variable" $z$ that captures the data generation process. In the main text, the encoder and decoder are linear transformations without bias terms, and the learnable bias is treated in Appendix B, which shows that the effect of the bias terms is equivalent to centering both the input and target to be zero-mean ($x \to x - \mathbb{E}[x]$, $y \to y - \mathbb{E}[y]$). Incorporating the bias terms thus does not affect the main results. Specifically, the encoder is defined as $z = W^\top x + \epsilon$, where $\epsilon \sim \mathcal{N}(0, \Sigma)$ is the noise distribution introduced by the reparameterization trick where the variance matrix $\Sigma = \text{diag}(\sigma_1^2, ..., \sigma_{d_1}^2)$ is assumed to be diagonal and independent from $x$. The decoder parametrizes the distribution $p(y|z) = \mathcal{N}(Uz, \eta_{\text{dec}}^2 I)$, where the variance $\eta_{\text{dec}}^2 I$ is to be isotropic and input-independent. In alignment with the standard practice, we also assume the prior distribution of latent variable $p(z) = \mathcal{N}(0, \eta_{\text{enc}}^2 I)$ is an isotropic normal distribution, and the encoding variances matrix $\Sigma$ is learned from the data distribution while $\eta_{\text{dec}}^2$ is not learnable. Lastly, we weigh the KL term by a coefficient $\beta$, which is a common practice in VAE training (Higgins et al., 2016). Hence, the objective of such a linear model reads,[1]

$$L_{\text{VAE}}(U, W, \Sigma) \tag{2}$$

$$= \mathbb{E}_x[-\mathbb{E}_{q(z|x)}\log(p(y|z)) + \beta D_{KL}(q(z|x)\|p(z; \eta_{\text{enc}}^2))] \tag{3}$$

$$= \frac{1}{2\eta_{\text{dec}}^2}\mathbb{E}_{x,\epsilon}\left[\|U(W^\top x + \epsilon) - y\|_2^2 + \beta\frac{\eta_{\text{dec}}^2}{\eta_{\text{enc}}^2}\|W^\top x\|^2\right] + \sum_{i=1}^{d_1}\frac{\beta}{2}\left(\frac{\sigma_i^2}{\eta_{\text{enc}}^2} - 1 - \log\frac{\sigma_i^2}{\eta_{\text{enc}}^2}\right) \tag{4}$$

$$= \frac{1}{2\eta_{\text{dec}}^2}\left[\mathbb{E}_x\|UW^\top x - y\|_2^2 + \text{Tr}(U\Sigma U^\top) + \underbrace{\beta\frac{\eta_{\text{dec}}^2}{\eta_{\text{enc}}^2}\text{Tr}(W^\top AW)}_{\ell_{mean}}\right] + \underbrace{\sum_{i=1}^{d_1}\frac{\beta}{2}\left(\frac{\sigma_i^2}{\eta_{\text{enc}}^2} - 1 - \log\frac{\sigma_i^2}{\eta_{\text{enc}}^2}\right)}_{\ell_{var}},$$

$$\tag{5}$$

where $A := \mathbb{E}_x[xx^\top]$ is the second moment of the input data. Note that a crucial feature of the KL term is that it decomposes into two terms, one that regularizes the variance of $z$ ($\ell_{var}$) and another that regularizes the mean of $z$ ($\ell_{mean}$). We will see that it is precisely the $\ell_{mean}$ term that causes the posterior collapse. Eq. (5) has ignored the partition function of the decoder because we treat $\eta_{\text{dec}}$ as a constant. We study the case of a learnable $\eta_{\text{dec}}$ in Section D. In comparison to the previous works (Lucas et al., 2019) that have treated the case of a learnable $\eta_{\text{dec}}$, our result is more general because our result also considers the effect of $\beta$ and allows for the case $d_1 \geq d_2$. It is also worth commenting on the difference between this setting and that of the pPCA setting (Nakajima et al., 2015): (1) the effect of $\beta$ is included in the VAE loss, (2) the prior of VAE is over the latent variable, whereas pPCA has it over the model parameters, and (3) the model can be overparametrized ($d_1 \geq d_0$).

**Notation**. To summarize, we use $x, y,$ and $z$ to denote the input variable, latent variable, and target variable, respectively. $\mathbb{E}_x$ denotes the expectation over the training set. $A := \mathbb{E}_x[xx^T]$ is the second moment matrix of the input $x$. $A$ is thus positive semidefinite by definition. The eigenvalue decomposition of $A$ is $A = P_A\Phi P_A^\top$, where $\Phi = \text{diag}(\phi_1, \cdots, \phi_{d_0})$ is the diagonal matrix for all $d_0 \leq D_0$ positive eigenvalues and $P_A = [p_1, \cdots, p_{d_0}]$ are matrices by concatenating $d_0$ eigenvectors $p_i \in \mathbb{R}^{D_0}$. $W$ and $U$ are learnable linear transformation matrices for the linear encoding and decoding processes. $\Sigma$ is the learnable diagonal latent variance matrix for encoder with diagonal entries $\sigma_i$. $\eta_{\text{enc}}$ is the standard deviation of the prior distribution $p(z)$. $\eta_{\text{dec}}$ is the standard deviation of decoded samples. A frequently used quantity is a whitened and rotated $x$: $\tilde{x} := \Phi^{-\frac{1}{2}}P_A^\top x$. Note that this transformation can be inverted: $x = P_A\Phi^{\frac{1}{2}}\tilde{x}$. We see that $\mathbb{E}_x\tilde{x}\tilde{x}^\top = I$. Furthermore, we define $Z := \mathbb{E}_{\tilde{x}}[y\tilde{x}^\top] = \mathbb{E}_x[yx^\top P_A\Phi^{-\frac{1}{2}}] \in \mathbb{R}^{d_2 \times d_0}$. Let $Z = F\Sigma_Z G^\top$ be the singular value decomposition of $Z$, where $F \in \mathbb{R}^{d_2 \times d_2}$ and $G \in \mathbb{R}^{d_0 \times d_0}$ are two orthogonal matrices. $\Sigma_Z \in \mathbb{R}^{d_2 \times d_0}$ is a rectangular diagonal matrix with $d^* = \min(d_0, d_2)$ singular values of $Z$ in the non-increasing order, i.e., $\zeta_1 \geq \zeta_2 \geq \cdots \geq \zeta_{d^*} \geq 0$.

---

[1]We note that $\mathbb{E}_x$ is the expectation over the training set. Also, we use the subscript "VAE" because the model can be seen as a conditional VAE, even though it may be more proper to call it a "general latent variable model."

# 4 Main Results

This section discusses the main results, whose proofs are presented in Appendix E. While $\Sigma$ is often a learnable parameter, we first assume that the KL term is sufficiently strong such that $\sigma_1 = \cdots = \sigma_{d_1} \approx \eta_{\text{enc}}$ is close to the prior value. We then compare with the case when it is learnable, and this comparison reveals that an optimizable $\sigma_i$ is not essential to the posterior collapse problem.

## 4.1 General Result

In this section, we prove two results that will be useful for understanding the nature of the VAE training objective and will be useful for us to find the global minimum. We first show that the VAE objective is equivalent to a matrix factorization problem with a special type of regularization.

**Proposition 1.** *Let* $\tilde{x} := \Phi^{-\frac{1}{2}} P_A^\top x$, $Z := E_{\tilde{x}}[y\tilde{x}^\top]$, *and*

$$(U^*, V^*) := \underset{(U,V)}{\arg\min}\, L(U, V) = \underset{(U,V)}{\arg\min} \|UV^\top - Z\|_F^2 + \text{Tr}(U\Sigma U^\top) + \beta \frac{\eta_{\text{dec}}^2}{\eta_{\text{enc}}^2} \|V\|_F^2. \tag{6}$$

*Given a fixed* $\Sigma$*, the minimizer of* $L_{\text{VAE}}(U, W, \Sigma)$ *is* $(U^*, W^*)$*, where* $W^*$ *is any solution of* $\Phi^{\frac{1}{2}} P_A^\top W = V^*$.

*Proof sketch.* The term $\ell_{var}$ is irrelevant to finding the optimal $U^*$ and $V^*$ when $\Sigma$ is fixed. Thus, the relevant objective can be obtained with the change of variables $\tilde{x} = \Phi^{-\frac{1}{2}} P_A^\top x$. □

The condition $\Phi^{\frac{1}{2}} P_A^\top W = V^*$ shows that when the data is low-rank, each solution $(U^*, V^*)$ corresponds to a manifold of solutions in the original parameter space. The effective loss $L(U, V)$ can be compared with the regularized singular value decomposition problem (Zheng et al., 2018). We see that the first term is the standard matrix factorization objective, while the second and third are unique regularization effects due to the VAE structure and the ELBO objective. In addition, the term $\Sigma$ in the second is the strength of the regularization for the norm of $U$, and a crucial difference with standard regularized matrix factorization is that $\Sigma$ is also a learnable matrix.

The next proposition finds, for any fixed $\Sigma$, the global minima $(U^*, V^*)$ of Eq. (6). In particular, the learning is characterized by the learning of the singular values of $U$ and $V$.

**Proposition 2.** *The optimal solution* $(U^*, V^*)$ *of* $\min_{U,V} L(U, V)$ *is given by*

$$U^* = F\Lambda P, \quad V^* = G\Theta P, \tag{7}$$

*where* $F \in \mathbb{R}^{d_2 \times d_2}$ *and* $G \in \mathbb{R}^{d_0 \times d_0}$ *are orthogonal matrices derived by the SVD of* $Z$*,* $P$ *is an arbitrary orthogonal matrix in* $\mathbb{R}^{d_1 \times d_1}$*, and* $\Lambda \in \mathbb{R}^{d_2 \times d_1}$ *and* $\Theta \in \mathbb{R}^{d_0 \times d_1}$ *are rectangular diagonal matrices with the diagonal elements*

$$\lambda_i = \sqrt{\max\left(0, \frac{\sqrt{\beta}\eta_{\text{dec}}}{\sigma_i \eta_{\text{enc}}}\left(\zeta_i - \frac{\sqrt{\beta}\sigma_i \eta_{\text{dec}}}{\eta_{\text{enc}}}\right)\right)}, \quad \theta_i = \sqrt{\max\left(0, \frac{\sigma_i \eta_{\text{enc}}}{\sqrt{\beta}\eta_{\text{dec}}}\left(\zeta_i - \frac{\sqrt{\beta}\sigma_i \eta_{\text{dec}}}{\eta_{\text{enc}}}\right)\right)}. \tag{8}$$

*For convention, we let* $\zeta_i = 0$ *when* $i > d^* = \min(d_0, d_2)$.

*Proof sketch.* The optimal $V^*$ is a function of $U$ under the zero gradient condition. Thus, the objective reduces to single-variate with respect to $U$. The optimal $U^*$ is constructed by its SVD $U = Q\Lambda P$, where the optimal $Q^*$ and $\Lambda^*$ can be determined given the SVD of $Z$, and $P$ is left as a free orthogonal matrix. $V^*$ is determined once $U^*$ is obtained. □

The readers are recommended to examine the form of the solutions closely. There are a few interesting features of the global minimum. One note that the sign of the term $\zeta_i - \sqrt{\beta}\sigma_i \eta_{\text{dec}}/\eta_{\text{enc}}$ is crucial, and can encourage the parameters $U$ and $V$ to be low-rank. Recall that $\sigma_i$ is the eigenvalue value of $ZZ^\top = E[y\tilde{x}^\top]E[y\tilde{x}^\top]^\top$, one can roughly identify $\zeta_i^2$ as the the strength of the alignment between the input $x$ and the target $y$. To see this, consider a simplified scenario where the target $y = \gamma M x$ is a linear function of the input, where $\gamma$ is the overall strength of the signal and $\|M\| = 1$ is a normalized orientation matrix, then

$$ZZ^\top = \gamma^2 M^\top A M, \tag{9}$$

which is a positive semidefinite matrix. We see that there are two distinctive sources of contribution to the magnitude of the eigenvalues of $ZZ^{\mathrm{T}}$. Its eigenvalues are large if either the overall strength $\gamma$ is large or if the orientation matrix $M$ aligns well with the covariance of the input feature $A$. Additionally, in the case of VAE, $\gamma M = I$, and $ZZ^{\mathrm{T}} = A$ is nothing but the covariance of input features, and $\zeta_i^2$ are the eigenvalues of $A$.

## 4.2  Linear VAE without Learnable $\Sigma$

We first consider the case where $\sigma_i$ is a constant that is completely determined by the prior: $\sigma_i = \eta_{\mathrm{enc}}$. This allows us to find a simplified form for the global minimum. The proof follows by plugging $\sigma_i = \eta_{\mathrm{enc}}$ into Proposition 2.

**Theorem 1.** *Let $\sigma_i = \eta_{\mathrm{enc}}$ for all $i$. Then, the global minimum has*

$$\lambda_i = \sqrt{\max\left(0, \frac{\sqrt{\beta}\eta_{\mathrm{dec}}}{\eta_{\mathrm{enc}}^2}\left(\zeta_i - \sqrt{\beta}\eta_{\mathrm{dec}}\right)\right)}, \quad \theta_i = \sqrt{\max\left(0, \frac{\eta_{\mathrm{enc}}^2}{\sqrt{\beta}\eta_{\mathrm{dec}}}\left(\zeta_i - \sqrt{\beta}\eta_{\mathrm{dec}}\right)\right)}. \quad (10)$$

There are three interesting observations of the global minimum. First of all, it depends crucially on the sign of $\zeta_i - \sqrt{\beta}\eta_{\mathrm{dec}}$ for all $i$. When the sign is negative for some $i$, the learned model becomes low-rank. Namely, some of the dimensions collapse with the prior. When the signs are all negative, we have a complete posterior collapse: both $U$ and $V$ are identically zero, so the latent variables have a distribution identical to the prior. A *complete* posterior collapse happens if and only if $\max_i \zeta_i - \sqrt{\beta}\eta_{\mathrm{dec}} \leq 0$. A *partial* posterior collapse happens if there exists $i$ such that $\zeta_i^2 - \sqrt{\beta}\eta_{\mathrm{dec}} \leq 0$. These two conditions give the precise conditions of posterior collapse in this scenario. This implies that having a sufficiently small $\beta$ will always prevent posterior collapse. The second observation is that the effect of $\beta$ is identical to that of $\eta_{\mathrm{dec}}$ because $\sqrt{\beta}$ and $\eta_{\mathrm{dec}}$ always appear together, and so one alternative way to fix posterior collapses is to use a sufficiently small $\eta_{\mathrm{dec}}$. From a Bayesian perspective, the latter method of tuning $\eta_{\mathrm{dec}}$ is better because $\eta_{\mathrm{dec}}$ comes directly from the (assumed) likelihood $p(x|\eta_{\mathrm{dec}})$. In contrast, the $\beta$ parameter is only an implementation technique that has obscure meaning in the Bayesian framework. Therefore, using a small $\eta_{\mathrm{dec}}$ can be a fix to the problem that is justified by the Bayesian principle. The third observation is that the condition for posterior collapse is completely independent of the parameter $\eta_{\mathrm{enc}}$, which is the desired variance according to the prior $p(z)$. This means that under a Gaussian assumption, the prior does not affect the posterior collapse at all.

Lastly, one also notices a potential problem. The eigenvalue of the second layer $U$ increases with $\sqrt{\beta}\eta_{\mathrm{dec}}$, while the first layer decreases with $\sqrt{\beta}\eta_{\mathrm{dec}}$, and so having a too-small $\beta$ or $\eta_{\mathrm{dec}}$ causes the model to have a very large norm, which can cause a significant problem for both empirical optimization and generalization. This problem is well-known in the studies about the use of $L_2$ regularization in deep learning: suppose we apply weight decay to two different layers of a ReLU net, and decrease the weight decay strength of one layer to zero, then the norm of this layer will tend to infinity, and the norm of the other layer will tend to zero (Mehta et al., 2018). However, in the next section, we will see that this problem is miraculously solved for VAE when $\sigma_i$ is learnable.

## 4.3  Linear VAE with Learnable $\Sigma$

Now, we consider the more general case of a learnable $\Sigma$. In practice, $\Sigma$ is often dependent on the input $x$. We make the simplification that $\Sigma$ is just a data-independent optimizable diagonal matrix, which is the common assumption in the related works (Lucas et al., 2019). In Section C, we consider the case when $\Sigma$ is data-dependent and show that our result remains unchanged. The following Corollary gives the optimal training objective as a function $\Sigma$ and is a direct consequence of proposition 2.

**Corollary 1.**

$$\min_{U,V} L(U,V) = \sum_{i=1}^{d_1} \zeta_i^2 - \left(\zeta_i - \frac{\sqrt{\beta}\sigma_i\eta_{\mathrm{dec}}}{\eta_{\mathrm{enc}}}\right)^2 \mathbb{1}_{\zeta_i > \frac{\sqrt{\beta}\sigma_i\eta_{\mathrm{dec}}}{\eta_{\mathrm{enc}}}} + \sum_{i=d_1+1}^{d^\star} \zeta_i^2, \quad (11)$$

*where the indicator $\mathbb{1}_{f>0} = 1$ when the corresponding inequality condition $f > 0$ is true, and $\mathbb{1}_{f>0} = 0$ otherwise.*

The constant term $\sum_{i=d_1+1}^{d^*} \zeta_i^2$ in Equation (11) only appears when the latent dimension $d_1$ is less than $d^* = \min(d_0, d_2)$. This is the common situation for VAE applications. It indicates that the model learns the large eigenvalues and ignores the small eigenvalues. This means that to find the optimal $\sigma_i$ of Eq. (5), one only has to find the global minimum of a reduced objective:

$$\min_{U,W} L_{\text{VAE}}(U, W, \Sigma) = \min_{U,V} \frac{1}{2\eta_{\text{dec}}^2} L(U, V) + \sum_{i=1}^{d_1} \frac{\beta}{2} \left( \frac{\sigma_i^2}{\eta_{\text{enc}}^2} - 1 - \log \frac{\sigma_i^2}{\eta_{\text{enc}}^2} \right) \tag{12}$$

$$= \frac{1}{2\eta_{\text{dec}}^2} \sum_{i=1}^{d_1} \left[ \underbrace{\zeta_i^2 - \left( \zeta_i - \frac{\sqrt{\beta}\sigma_i\eta_{\text{dec}}}{\eta_{\text{enc}}} \right)^2 \mathbb{1}_{\zeta_i > \frac{\sqrt{\beta}\sigma_i\eta_{\text{dec}}}{\eta_{\text{enc}}}} + \beta\eta_{\text{dec}}^2 \left( \frac{\sigma_i^2}{\eta_{\text{enc}}^2} - 1 - \log \frac{\sigma_i^2}{\eta_{\text{enc}}^2} \right)}_{:=l_i(\sigma_i)} \right] + constant. \tag{13}$$

The optimal $\sigma_i^*$ can thus be obtained by minimizing each $l_i$ independently: $\sigma_i^* = \arg\min_{\sigma>0} l_i(\sigma)$.

**Proposition 3.** *The optimal $\sigma_i^*$ of $l_i(\sigma)$ is*

$$\sigma_i^* = \begin{cases} \frac{\sqrt{\beta}\eta_{\text{dec}}}{\zeta_i} \eta_{\text{enc}} & if\ \beta\eta_{dec}^2 < \zeta_i^2; \\ \eta_{\text{enc}} & if\ \beta\eta_{dec}^2 \geq \zeta_i^2. \end{cases} \tag{14}$$

This proposition gives an explicit expression for $\sigma_i^*$. On the one hand, we see that there is a threshold value for $\beta$. If $\beta$ is sufficiently large, $\sigma_i$ will be identical to the prior value $\eta_{\text{enc}}$, in agreement with our assumption in the previous section. On the other hand, the learned variance $\sigma_i^*$ is a function of $\beta$ if $\beta$ is below a threshold. We will see that this threshold is the necessary and sufficient condition for posterior collapse to happen in a learnable $\Sigma$ setting. Thus, the learned variance being identical to the prior variance is also a signature of posterior collapse. The following theorem gives the precise form of the global minimum.

**Theorem 2.** *The global minimum of $L_{\text{VAE}}(U, W, \Sigma)$ is given by*

$$U^* = F\Lambda P, \tag{15}$$

*$W^*$ is the solution of*

$$\Phi^{\frac{1}{2}} P_A^\top W = G\Theta P, \tag{16}$$

*where $F$ and $G$ are derived by the SVD of $Z$, $P$ is an arbitrary orthogonal matrix in $\mathbb{R}^{d_1 \times d_1}$, and $\Lambda = \text{diag}(\lambda_1, ..., \lambda_{d_1})$ and $\Theta = \text{diag}(\theta_1, ..., \theta_{d_1})$ are diagonal matrices such that*

$$\lambda_i = \frac{1}{\eta_{\text{enc}}} \sqrt{\max\left(0, \zeta_i^2 - \beta\eta_{\text{dec}}^2\right)}; \tag{17}$$

$$\theta_i \begin{cases} = \frac{\eta_{\text{enc}}}{\zeta_i} \sqrt{\max\left(0, \zeta_i^2 - \beta\eta_{\text{dec}}^2\right)} & when\ \zeta_i^2 > 0; \\ = 0 & otherwise. \end{cases} \tag{18}$$

*The optimal $\Sigma^* = \text{diag}(\sigma_1^{*2}, ..., \sigma_{d_1}^{*2})$ such that*

$$\sigma_i^* = \begin{cases} \frac{\sqrt{\beta}\eta_{\text{dec}}}{\zeta_i} \eta_{\text{enc}} & \beta\eta_{dec}^2 < \zeta_i^2, \\ \eta_{\text{enc}} & \beta\eta_{dec}^2 \geq \zeta_i^. \end{cases} \tag{19}$$

*for $i \leq \min(d_0, d_2)$. For $i > \min(d_0, d_2)$, $\sigma_i^* = 0$.*

*Proof.* The optimal solution $U^*, W^*, \Sigma^*$ are obtained by combining proposition 1, 2, and 3. $\square$

Comparing with the solution in section 4.2, one notices two things: (a) the conditions for complete or partial posterior collapse remain unchanged, which implies that a learnable latent variance is neither qualitatively nor quantitatively relevant for the posterior collapse problem even though the functional form of the eigenvalues changed; (b) the magnitude of each of the two layers no longer scales with $\sqrt{\beta}\eta_{\text{dec}}$, and so using a small $\beta$ or $\eta$ will not directly cause the model to diverge in the

norm, which suggests using that making $\Sigma$ learnable can have the unexpected practical advantage of stabilizing the training.

Additionally, one also notices that $\beta\eta_{\text{dec}}^2$ has the effect of keeping the learned model low-rank by removing all the eigenvalues of the learned model below it. This can be directly compared with the effect of using a latent dimension smaller than the input dimension: $d_1 < d_0$. In the latter case, the smallest $d_1 - d_0$ singular values are also pruned. There is a difference between the two types of low-rankness: using a large $\beta\eta_{\text{dec}}^2$ both removes all the singular values below it and shrinks the remaining ones while using a small latent dimension only removes the smaller singular values without affecting the rest. This is similar to the difference between soft thresholding estimation and hard thresholding estimation in statistics (Wasserman, 2013). This suggests that partial posterior collapses are not necessarily undesirable because, during a partial posterior collapse, the latent variable models automatically perform a degree of sparse learning, which is theoretically understood to help denoising the signal and lead to better generalization (Markovsky, 2012). That being said, complete posterior collapse should always be avoided.

### 4.4 Learnable $\eta_{\text{dec}}$

Our result can also be extended to the case when $\eta_{\text{dec}}$ is learnable, which has been suggested by Lucas et al. (2019) as a remedy for the posterior collapse. To do this, we need to include the partition function of the decoder, proportional to $\log\eta_{\text{dec}}^2$, that has been ignored in Eq. (5). We present the detailed analysis in Section D. Our analysis shows that even if $\eta_{\text{dec}}$ is learnable, posterior collapse can happen for some datasets. In addition to the fact that it is also possible for collapses not to happen when $\eta_{\text{dec}}$ is not learned, we conclude that $\eta_{\text{dec}}$ does not have a causal relation with posterior collapse. Our analysis also suggests a way to fix posterior collapse for VAE: make $\eta_{\text{dec}}$ learnable and set $\beta < d_2/d^*$. Note that the condition $\beta < d_2/d^*$ is tight in the sense that if it does not hold, then there exists a data distribution such that complete collapse can happen. This condition also highlights that it is important to introduce the $\beta$ coefficient for VAEs because, for VAE, $d_2/d^* = 1$, and this condition translates to $\beta < 1$; namely, vanilla VAE cannot avoid complete collapse.

Our result also implies that $\beta$ has a highly nonlinear effect on learning depending on the architecture. For example, when the model is underparameterized ($d_1 < d^*$), using a small $\beta$ does not cause any problem, whereas for an overparametrized model, a small $\beta$ causes the decoder variance to converge towards 0.

### 4.5 Implications

Our main results have implications for both the problem of posterior collapse and the practice of latent variable models in general.

**The cause of posterior collapse**. One important implication is the identification of the cause of the posterior collapse problem and the potential ways to fix it. Our results suggest that

- a learnable (data-dependent or not) latent variance is not the cause of posterior collapse;
- changing the variance of the prior cannot fix or influence the posterior collapse problem;
- comparing with the results in Lucas et al. (2019), $\eta_{\text{dec}}$ being learnable or not is causally related to the posterior collapse problem;
- the values of $\eta_{\text{dec}}$ and $\beta$ are crucial for the posterior collapse;
- choosing appropriate $\beta$ is still needed: a sufficiently small $\beta$ can avoid posterior collapse.

Note that the effect of a small $\beta$ (large $\eta_{\text{dec}}$) weakens the prior (reconstruction) term, and so the cause of the posterior collapse must be the competition between the prior term, which regularizes the complexity of the model, and the likelihood term, which encourages accurate recognition/reconstruction. Our results suggest that one can ignore the effect of the $\ell_{var}$ term in studying the mechanism of posterior collapse. Ignoring the $\ell_{var}$ Eq. 5, one sees that the posterior collapse is caused by the competition between the likelihood and $\ell_{mean}$, which is precisely the regularization effect on the mean of $z$.

There is an interesting alternative perspective on the nature of the posterior from the viewpoint of the loss landscape geometry. The following theorem states that the origin (where all parameters are zero) is either a saddle or the global minimum for this problem. Since we have shown that $\sigma_i$ does not affect the collapse, we simply let $\sigma_i = \eta_{\text{enc}}$ as in Section 4.2.

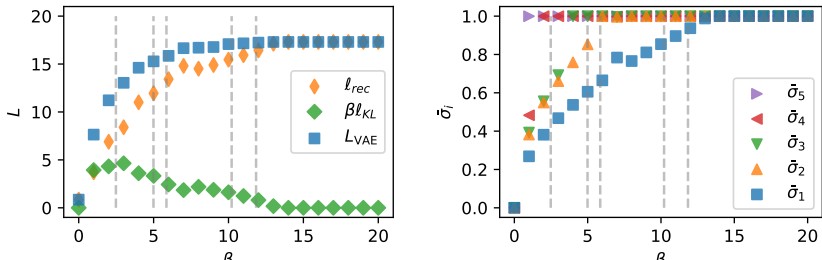

Figure 1: Training loss $L$ and $\bar{\sigma}_i$ versus $\beta$ on synthetic regression dataset. $\bar{\sigma}_i$ is measured by averaging over the training set. The vertical dashed lines show where the theory predicts a partial collapse. Complete posterior collapse happens at roughly $\beta = 14$.

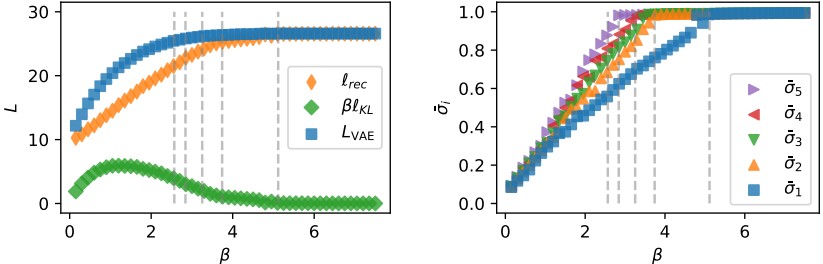

Figure 2: Training loss and $\bar{\sigma}_i$ versus $\beta$ for MNIST dataset. The vertical dashed lines show where the theory predicts a partial collapse. The posterior collapse happens for the MNIST dataset at around $\beta = 5$.

**Theorem 3.** *The Hessian of Eq. 5 at* $0$ *is positive semidefinite if and only if it is the global minimum.*

The surprising aspect is that for the latent variable model, there is no intermediate case where the origin is a local minimum but not global. Therefore, the origin is, in fact, a very special point in the landscape of a latent variable model, in the sense that a key global property of the landscape (namely, the global minimum) is determined by the local geometry of the model at the origin. Noting that our model can be seen as a direct generalization of the Bayesian linear regression to a deeper architecture, it also becomes reasonable to suspect that the posterior collapse problem is a unique problem of deep learning because the standard Bayesian linear regression does not suffer posterior collapse because the origin can never be a local maximum of the posterior (Bishop and Nasrabadi, 2006). Dai et al. (2020) also finds the origin to be a very special point in a general deep nonlinear VAE structure and that it can be a local minimum under various settings. However, the implication of our work is broader. The origin is not only a special point for the autoencoding model families but can actually be a special point for a very broad of model classes (namely, the model class of general latent variable models). The problem of posterior collapse is thus not limited to autoencoders but can also be relevant to common regression and classification tasks.

**Connection to other types of collapses**. Our result suggests that there are some interesting connections between the posterior collapse phenomenon and the neural collapse phenomenon in supervised learning (Papyan et al., 2020) and dimensional collapse phenomenon in self-supervised learning (SSL) (Jing et al., 2021). Ziyin et al. (2022a) and Ziyin and Ueda (2022) shows that the neural collapse phenomenon for a two-layer model can be understood through the change of the stability at the origin, which is determined by the competition between the signal strength ($\mathbb{E}[xy]$) of the data distribution and the regularization strength of weight decay. For SSL, Ziyin et al. (2022b) also shows that the stability of the origin is important and that it is decided by the competition between the level of data variation and the data augmentation strength. Our result suggests that the posterior collapse problem can also be understood through the stability at the origin. This might imply that there could be some universal cause of all these collapses that have been discovered independently in different subfields of deep learning, and one important future direction would be to study these phenomena from a unified perspective.

**Insights for latent variable model practices**. While we have primarily focused on discussing the phenomena of posterior collapse, our results also shed light on latent variable models (including VAE) in practice when there is no complete posterior collapse. Specifically, our results suggest that

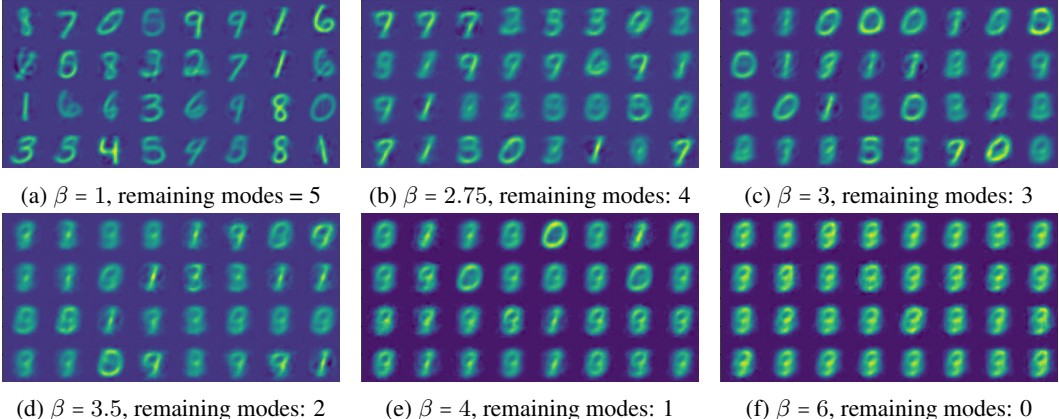

(a) $\beta = 1$, remaining modes = 5     (b) $\beta = 2.75$, remaining modes: 4     (c) $\beta = 3$, remaining modes: 3

(d) $\beta = 3.5$, remaining modes: 2     (e) $\beta = 4$, remaining modes: 1     (f) $\beta = 6$, remaining modes: 0

Figure 3: MNIST generation under different $\beta$. The generated images lose diversity and variation as $\beta$ increases. The number of modes left is estimated by the theoretical prediction of thresholds of each singular value.

- latent variable models perform sparse learning through soft thresholding or hard thresholding or both;
- thus, *partial* posterior collapse may actually be desirable;
- making the latent variance learnable can help stabilize training and avoid divergence of model parameters;
- when $\eta_{\mathrm{dec}}$ is not learned, the effect of increasing $\beta$ is identical to the effect of decreasing $\eta_{\mathrm{dec}}$;
- when $\eta_{\mathrm{dec}}$ is learned, one needs to pay special care to choose a suitable $\beta$.

## 5   Numerical Examples

This section empirically examines our theoretical claims for linear models and demonstrates that our key theoretical insights generalize well to nonlinear models and natural data.

*Setting.* We illustrate our results on both synthetic data and natural data. For synthetic data, we sample input data $x$ from multivariate normal distribution $\mathcal{N}(0, A)$, and target data $y = Mx$ is obtained by a linear transformation. Specifically, we choose $d_0 = d_2 = 5$. As an example of natural data, we also experiment with the standard MNIST data. Following common practices, we choose $\eta_{\mathrm{dec}} = \eta_{\mathrm{enc}} = 1$. For non-linear VAE models, we consider two-layer fully connected neural networks for the encoder and decoder with both ReLU and Tanh activation functions and with hidden dimension $d_h$. For synthetic dataset $d_h = 8$, and $d_h = 2048$ for real-world data. In contrast to our assumption that the variances $\Sigma$ of encoded $z$ are independent from the input $x$, we parameterize the variance of each encoded $z$ by a linear transformation or a two-layer neural network, i.e., $\Sigma(x) = [\mathrm{Linear/MLP}](x)$. This data-dependent modeling is closer to the common practice, and the comparison can justify the correctness of our theory. The model is optimized by Adam with a learning rate of $10^{-3}$. The results are reported after the convergence. For MNIST, the learning rate is $10^{-4}$.

*Results.* Linear models are found to agree precisely with the theoretical results, so we only present the results in the appendix. We focus on exploring the nonlinear models in the main text. We first consider a simple regression task with MLP encoder and decoders with the ReLU activation (Figure 1). Here, we see that the theoretical prediction of loss function $L_{\mathrm{VAE}}$ agrees well with empirical observation. Moreover, the threshold of complete posterior collapse is also perfectly predicted. For completeness, we also present the case when (1) the activation is Tanh in Appendix A.1. We note that the results are similar. The observation is similar to the standard MNIST dataset with a nonlinear encoder and decoder. See Figure 2. For illustration, we also present the generated MNIST images by non-linear $\beta$-VAE trained with different choices of $\beta$ in Figure 3. The latent dimension is five as described before. When there are 5 non-collapsed modes, the generated images are both sharp and contain meaningful variations. As the number of remaining non-collapsed modes reduces to zero, we see that the generated images become increasingly blurred, and the variation between the data also diminishes. When the model completely collapses, the model outputs a constant, as the theory

suggests. Moreover, we note that the values of $\beta$ are chosen according to the theoretical thresholds for each mode to collapse, i.e, the top-5 $\zeta_i$ are $[5.12, 3.74, 3.25, 2.84, 2.57]$. We see that the theoretical thresholds provide good predictive power for the behavior of mode collapse qualitatively.

## 6 Outlook

In this work, we have tackled the problem of posterior collapse from a loss landscape point of view. Our work also contributes to the fundamental theory of deep learning. The linear VAE architecture can be seen as a deep linear model with two layers, whose loss landscape is highly nontrivial. In this perspective, our results advance those results in Ziyin et al. (2022a), where the dimension of the output space is limited to 1d. The limitation of our work is obvious: our theory only deals with the landscape, and it is unclear how the dynamics of gradient-based methods could contribute to the collapse problem. In fact, there is strong evidence that stochastic gradient descent can bias the model towards low-rank or sparse solutions (Arora et al., 2019; Ziyin et al., 2021), and, in the context of posterior collapse, these are precisely the collapsed solutions. One important future direction is thus to study the role of dynamics in influencing posterior collapse.

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
