# A   Additional Experiments

## A.1   Regression

Figure 4 shows the results of a linear model in the regression setting. Figure 5 shows the performance of Tanh MLP in the regression setting. The complete posterior collapse is well predicted by our theory.

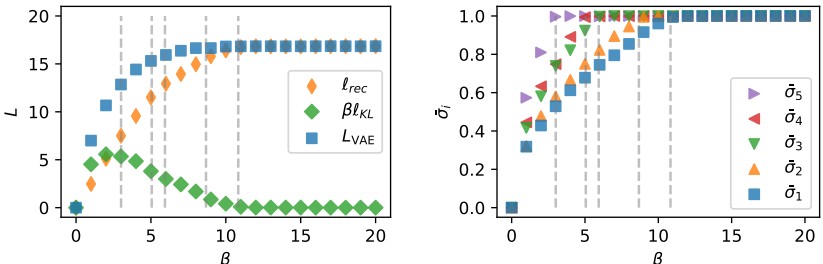

Figure 4: Training loss $L$ and $\sigma_i$ versus $\beta$ for linear regression. The theoretical prediction is plotted as vertical dashed lines.

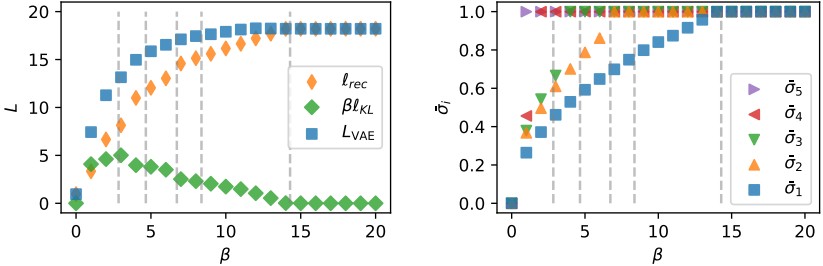

Figure 5: Training loss $L$ and $\sigma_i$ versus $\beta$ for MLP encoder and decoder with Tanh activation function. The theoretical prediction is plotted as vertical dashed lines.

## B  Effect of Bias

Here, we study a general linear encoding and decoding model equipped with a bias term. Following the previous notation, the encoder is $z = W^\top x + b_e + \epsilon$ and the decoded distribution is $p(y|z) = \mathcal{N}(Uz + b_d, \eta_{\text{dec}}^2 I)$. Then, the objective of general VAE reads

$$L_{\text{VAE}}(W, U, b_e, b_d, \Sigma) = \frac{1}{2\eta_{\text{dec}}^2} \mathbb{E}_{x,\epsilon} \left[ \| U(W^\top x + b_e + \epsilon) + b_d - y \|^2 + \beta \frac{\eta_{\text{dec}}^2}{\eta_{\text{enc}}^2} \| W^\top x + b_e \|^2 \right] \quad (20)$$

$$+ \sum_{i=1}^{d_1} \frac{\beta}{2} \left( \frac{\sigma_i^2}{\eta_{\text{enc}}^2} - 1 - \log \frac{\sigma_i^2}{\eta_{\text{enc}}^2} \right). \quad (21)$$

One can show that at optima, the learned biases must take the following form.

**Proposition 4.** *The optimal biases are $b_e^* = -W^\top \mathbb{E}_x[x]$ and $b_d^* = \mathbb{E}_x[y]$.*

*Proof.* The gradient of $L_{\text{VAE}}$ with respect to $b_e$ and $b_d$ are zero when $b_e$ and $b_d$ are optimal. That is,

$$\frac{\partial L_{\text{VAE}}}{\partial b_e} = \frac{1}{\eta_{\text{dec}}^2} \mathbb{E}_{x,\epsilon} \left[ U^\top (U(W^\top x + b_e + \epsilon) + b_d - y) + \beta \frac{\eta_{\text{dec}}^2}{\eta_{\text{enc}}^2} (W^\top x + b_e) \right] \quad (22)$$

$$= \frac{1}{\eta_{\text{dec}}^2} \left[ \left( U^\top U + \beta \frac{\eta_{\text{dec}}^2}{\eta_{\text{enc}}^2} I \right) (W^\top \mathbb{E}_x x + b_e) + U^\top (b_d - \mathbb{E}_x y) \right] = 0, \quad (23)$$

and,

$$\frac{\partial L_{\text{VAE}}}{\partial b_d} = \frac{1}{\eta_{\text{dec}}^2} \mathbb{E}_{x,\epsilon} (U(W^\top x + b_e + \epsilon) + b_d - y) \quad (24)$$

$$= \frac{1}{\eta_{\text{dec}}^2} \left[ U(W^\top \mathbb{E}_x x + b_e) + (b_d - \mathbb{E}_x y) \right] = 0. \quad (25)$$

Those condition holds if and only if $b_e^* = -W^\top \mathbb{E}_x x$ and $b_d^* = \mathbb{E}_x y$. $\qquad \square$

In particular, this means that the effect of a learnable encoder bias is the same as a data-preprocessing scheme of making $x$ zero-mean. The effect of a learnable decoder bias is the same as a data-preprocessing scheme of making $y$ zero-mean.

## C  Case of a Data-Dependent Encoding Variance

For completeness, we extend the result in Section 4.3 and consider the case when the learnable variance of the latent variable $z$ is $x$-dependent, which is common in practice. Meanwhile, one might also consider the case when the variance in the decoder is learned: for concision, we do not consider this case because it is rather rare in practice.

In the same spirit, we consider the simplest case of a data-dependent variance, where the standard derivation of $z$ linearly depends on $x$. We will see that in this case, the system is no longer analytically solvable. The standard deviation is

$$\sigma(x) = \text{diag}(|Cx + f|) := \text{diag}(\sigma_1(x), ..., \sigma_{d_1}(x)), \quad (26)$$

where $C \in \mathbb{R}^{d_1 \times d_0}$ and $f \in \mathbb{R}^{d_1}$ are the learnable parameters. The latent variable $z$ is thus generated by

$$z = Wx + \sigma(x)\epsilon = Wx + \text{diag}(Cx + f)\epsilon, \quad (27)$$

where $\epsilon \sim \mathcal{N}(0, I_{d_1})$. To emphasize the important terms, we further assume that $x$ is zero-mean: $\mathbb{E}[x] = 0$.[2]

---

[2] As we have shown, this can be precisely achieved when the encoder and encoder have a learnable bias.

Using this definition of $\sigma$ in Eq. (4), one obtains the following objective with Data-Dependent (encoding) Variance (DDV):

$$L_{\mathrm{VAE}}^{\mathrm{DDV}}(U, W, C, f) \tag{28}$$

$$= \frac{1}{2\eta_{\mathrm{dec}}^2} \mathbb{E}_{x,\epsilon}\left[ \|(UW^\top x - y) + U\sigma(x)\epsilon\|^2 + \beta \frac{\eta_{\mathrm{dec}}^2}{\eta_{\mathrm{enc}}^2} \|W^\top x\|^2 \right]$$

$$+ \frac{\beta}{2} \sum_{i=1}^{d_1} \mathbb{E}_x \left( \frac{\sigma_i^2(x)}{\eta_{\mathrm{enc}}^2} - 1 - \log \frac{\sigma_i^2(x)}{\eta_{\mathrm{enc}}^2} \right) \tag{29}$$

$$= \frac{1}{2\eta_{\mathrm{dec}}^2} \mathbb{E}_{x,\epsilon}\left[ \|UW^\top x - y\|^2 + \mathrm{Tr}(U\sigma(x)\epsilon\epsilon^\top\sigma(x)U^\top) + 2\mathrm{Tr}(U(W^\top x - y)\epsilon^\top\sigma(x)U^\top) \right.$$

$$\left. + \beta \frac{\eta_{\mathrm{dec}}^2}{\eta_{\mathrm{enc}}^2} \|W^\top x\|^2 \right] + \frac{\beta}{2} \sum_{i=1}^{d_1} \mathbb{E}_x \left( \frac{\sigma_i^2(x)}{\eta_{\mathrm{enc}}^2} - 1 - \log \frac{\sigma_i^2(x)}{\eta_{\mathrm{enc}}^2} \right). \tag{30}$$

The relevant expectation values can be computed easily:

$$\mathbb{E}_x \sigma_i^2(x) = f_i^2 + C_{i\cdot} A C_{i\cdot}^\top \coloneqq \Sigma_i^{\mathrm{DDV}} \tag{31}$$

$$\mathbb{E}_{x,\epsilon} \mathrm{Tr}(U\sigma(x)\epsilon\epsilon^\top\sigma(x)U^\top) = \mathrm{Tr}(U\,\mathrm{diag}(\mathbb{E}_x\sigma_1^2(x), \cdots, \mathbb{E}_x\sigma_{d_1}^2(x))U^\top) \tag{32}$$

$$\mathbb{E}_{x,\epsilon} \mathrm{Tr}(U(W^\top x - y)\epsilon^\top\sigma(x)U^\top) = \mathbb{E}_x \mathrm{Tr}(U(W^\top x - y)\mathbb{E}_\epsilon\epsilon^\top\sigma(x)U^\top) = 0, \tag{33}$$

where $\mu_x$ is the mean vector of input variable $x$, and $f_i C_{i\cdot} \mu_x$ is the inner product of the $i$-th row of $C$ and $\mu_x$ multiplied by a scalar $f_i$. This corollary means that the loss function can be written in the following form:

$$L_{\mathrm{VAE}}^{\mathrm{DDV}}(U, W, C, f) = \frac{1}{2\eta_{\mathrm{dec}}^2} \mathbb{E}_{x,\epsilon}\left[ \|UW^\top x - y\|^2 + \mathrm{Tr}(U\Sigma^{\mathrm{DDV}}U^\top) + \beta \frac{\eta_{\mathrm{dec}}^2}{\eta_{\mathrm{enc}}^2} \|W^\top x\|^2 \right]$$

$$+ \frac{\beta}{2} \sum_{i=1}^{d_1} \mathbb{E}_x \left( \frac{\sigma_i^2(x)}{\eta_{\mathrm{enc}}^2} - 1 - \log \frac{\sigma_i^2(x)}{\eta_{\mathrm{enc}}^2} \right). \tag{34}$$

What makes the problem analytical intractable is the term $\mathbb{E}_x \log(\sigma_i^2(x))$. However, we can still obtain some very insightful qualitative results from it.

The following lemma will help us show that it is always better to have $C = 0$.

**Lemma 1.** *For any $C$, $f$, there exists $f'$ such that $\mathbb{E}_x \sigma_i^2(x; C, f) = \mathbb{E}_x \sigma_i^2(x; C = 0, f')$.*[3]

*Proof.* By definition,

$$\mathbb{E}_x \sigma_i^2(x; C, f) = f_i^2 + C_{i\cdot} A C_{i\cdot}^\top, \tag{35}$$

and

$$\mathbb{E}_x \sigma_i^2(x; C = 0, f') = (f_i')^2. \tag{36}$$

Now, setting

$$f_i' = \sqrt{f_i^2 + C_{i\cdot} A C_{i\cdot}^\top} \tag{37}$$

is sufficient to make the two equal. $\square$

We can now prove that it is always better to have $C = 0$.

**Proposition 5.** *For any $U$, $W$, $C, f$, there exists $f'$ such that*

$$L_{\mathrm{VAE}}^{\mathrm{DDV}}(U, W, C, f) \geq L_{\mathrm{VAE}}^{\mathrm{DDV}}(U, W, 0, f'). \tag{38}$$

*Proof.* Throughout, we let $f'$ equal to the form given by Lemma 1.

By Eq. (31), the loss function can be written as the sum of a term $L_0$ that depends only on $U$, $W$ and $\Sigma^{\mathrm{DDV}}$ and the logarithmic term:

$$L_{\mathrm{VAE}}^{\mathrm{DDV}}(U, W, C, f) = L_1(U, W, \Sigma^{\mathrm{DDV}}) - \frac{\beta}{2} \mathbb{E}_x \log \sigma_i^2(x). \tag{39}$$

---

[3]Note that we have now explicitly written out $C$ and $f$ to emphasize that $\sigma$ is also a function of $C$ and $f$

However, by Lemma (1), we have

$$L_{\text{VAE}}^{\text{DDV}}(U, W, 0, f') = L_1(U, W, \Sigma^{\text{DDV}}) - \frac{\beta}{2} \log \mathbb{E}_x \sigma_i^2(x). \tag{40}$$

Noting that $-\log \sigma_i^2(x)$ is convex, we have, for any $C$ and $f$,

$$-\mathbb{E}_x \log \sigma_i^2(x) \geq -\log \mathbb{E}_x \sigma_i^2(x). \tag{41}$$

This implies that

$$L_{\text{VAE}}^{\text{DDV}}(U, W, C, f) - L_{\text{VAE}}^{\text{DDV}}(U, W, 0, f') = -\mathbb{E}_x \log \sigma_i^2(x) + \log \mathbb{E}_x \sigma_i^2(x) \geq 0. \tag{42}$$

This completes the proof. □

When $C = 0$, the encoder variance becomes data-independent, and the global minimum is thus, again, given by the main results in the main text. This result shows that a learnable data-dependent encoder variance does not have any quantitative difference at the global minimum when compared with the case of a data-independent encoder variance. This result is directly supported by our numerical results in Section 5, where the experiments are done for the case where the encoder variances are actually learned.

## D  Case of Learnable Decoding Variance $\eta_{\text{dec}}^2$

We first give an explicit form of the loss function at the global minimum found in Theorem 2. Using the optimal $U^*, W^*, \Sigma^*$, the analytical formulation of the minimal $L_{\text{VAE}}$ can be obtained.

**Corollary 2.** *The minimal value of the objective function $L_{\text{VAE}}$ is*

$$\min_{U,W,\Sigma} L_{\text{VAE}}(U, W, \Sigma) = \frac{1}{2\eta_{\text{dec}}^2} \left[ \sum_{i=1}^{d^*} \zeta_i^2 - \sum_{i:\zeta_i^2 > \beta \eta_{\text{dec}}^2}^{d_1} \zeta_i^2 \left( 1 + \frac{\beta \eta_{\text{dec}}^2}{\zeta_i^2} \left( \log \frac{\beta \eta_{\text{dec}}^2}{\zeta_i^2} - 1 \right) \right) \right], \tag{43}$$

*where $\zeta_i^2$ are sorted in non-increasing order. For convenience, we let $\zeta_i^2 = 0$ for $d^* \leq i \leq d_1$ when $d_1 > d^*$.*

Corollary 2 gives the global minimum of the objective for a fixed decoding variance $\eta_{\text{dec}}^2$. The first summation considers all eigenvalues $\zeta_i^2$ while the second summation considers non-zero first $d_1$ eigenvalues.

Here, we discuss the VAE with a Learnable Decoding Variance (LDV) $\eta_{\text{dec}}^2$. For shorthand, we denote $\eta_{\text{dec}}^2 := s \in (0, \infty)$. When we want to optimize over $s$, we also need to include the partition function, $\frac{d_2}{2} \log s$, of the decoder in the loss $L_{\text{VAE}}^{\text{LDV}}$. We note that this partition function has been ignored in the main text because $s$ has been treated as a constant for $L_{\text{VAE}}$. The loss function $L_{\text{VAE}}^{\text{LDV}}$ with the optimal $U^*, W^*$, and $\Sigma^*$ is thus given by combining Eq. (43) and the partition function $\frac{d_2}{2} \log s$:

$$G(s) := L_{\text{VAE}}^{\text{LDV}}(U = U^*, W = W^*, \Sigma = \Sigma^*, \eta_{\text{dec}}^2 = s)$$

$$= \frac{1}{2s} \sum_{i=1}^{d^*} \zeta_i^2 - \frac{1}{2} \sum_{i:\zeta_i^2 > \beta s}^{d_1} \left[ \frac{\zeta_i^2}{s} + \beta \left( \log \frac{\beta s}{\zeta_i^2} - 1 \right) \right] + \frac{d_2}{2} \log s. \tag{44}$$

Next, we investigate how $\beta$ affects the learnable decoding variance $s$ and identify the optimal $s^*$ under various conditions. Then, we show that, even with a learnable $\eta_{\text{dec}}^2$, the specific choice of $\beta$ can lead to or avoid the posterior collapse.

Moreover, for clarity, let $\hat{d}^*$ be the number of non-zero $\zeta_i^2$ for $1 \leq i \leq d^*$, and $\hat{d}_1$ be the number of non-zero $\zeta_i^2$ for $1 \leq i \leq d_1$. It is easy to see that $\hat{d}_1 \leq \hat{d}^*$. The loss is

$$G(s) = \frac{1}{2s} \sum_{i=1}^{\hat{d}^*} \zeta_i^2 - \frac{1}{2} \sum_{i}^{\hat{d}_1} F_i(s) + \frac{d_2}{2} \log s, \tag{45}$$

where

$$F_i(s) := \begin{cases} \frac{\zeta_i^2}{s} + \beta \left( \log \frac{\beta s}{\zeta_i^2} - 1 \right) & \text{when } s < \frac{\zeta_i^2}{\beta}, \\ 0 & \text{otherwise.} \end{cases} \tag{46}$$

**Lemma 2.** $G(s)$ *is differentiable.*

*Proof.* It suffices to check that $F_i(s)$ is differentiable on $(0, \infty)$ for all $i$. By definition, $F_i(s)$ is differentiable except at $\zeta_i^2 = \beta s$, and thus it suffices to check its differentiability at $\zeta_i^2/\beta$.

First of all, $F$ is continuous:

$$\lim_{s \to (\zeta_i^2/\beta)^-} F(s) = 0 = \lim_{s \to (\zeta_i^2/\beta)^+} F(s). \tag{47}$$

Then, $F$ is differentiable:

$$\lim_{s \to (\zeta_i^2/\beta)^-} F_i'(s) = \lim_{s \to (\zeta_i^2/\beta)^-} \frac{\beta}{s^2}(s - \zeta_i^2/\beta) = 0 = \lim_{s \to (\zeta_i^2/\beta)^+} F_i'(s). \tag{48}$$

This finishes the proof. □

Therefore, we only need to check the stationary points and the right limit of $G(s)$ at $0$ and the left limit at $\infty$. We proceed by first considering the monotonicity over intervals defined by the piecewise function and then narrowing down the solution of $G'(s) = 0$ into a specific interval.

Let $s_p := \frac{1}{\beta}\zeta_p^2$ for $p = 1, \cdots, \hat{d}_1$. We define $s_{\hat{d}_1 + 1} = 0$ and $s_0 = \infty$. Because the $\zeta_i$ are listed in non-increasing order, we have $0 = s_{\hat{d}_1 + 1} < s_{\hat{d}_1} \leq \cdots \leq s_1 < s_0 = +\infty$. Then $(0, +\infty) = \bigcup_{p=0}^{\hat{d}_1}[s_{p+1}, s_p) - \{0\}$ can be decomposed into the union of a set of intervals. For each interval $[s_{p+1}, s_p)$,

$$G'(s) = \frac{1}{2s^2}\left[(d_2 - \beta p)s - \sum_{i=p+1}^{d^*} \zeta_i^2\right], \tag{49}$$

where we implicitly define $\sum_{i=p+1}^{\hat{d}^*} \zeta_i^2 := 0$ when $p \geq \hat{d}^*$.

The following lemma states the number of stationary point of $G(s)$ in an interval $[s_{p+1}, s_p)$.

**Lemma 3.** *At each interval $[s_{p+1}, s_p), 0 \leq p \leq \hat{d}_1$, $G'(s)$ has at most one stationary point when $(d_2 - \beta\hat{d}_1) \neq 0$ or infinite stationary points when $(d_2 - \beta\hat{d}_1) = 0$ and $p = \hat{d}_1 = \hat{d}^*$.*

*Proof.* The existence of stationary points requires $G'(s) = 0$, which is equivalent to $(d_2 - \beta p)s = \sum_{i=p+1}^{d^*} \zeta_i^2$. When $p < \hat{d}_1$, $\sum_{i=p+1}^{d^*} \zeta_i^2 > 0$ holds. Therefore, $G'(s) = 0$ has at most one solution.

When $p = \hat{d}_1 = \hat{d}^*$, $G'(s) = 0$ holds only if $(d_2 - \beta\hat{d}_1) = 0$. Then, $\forall s \in (0, s_{\hat{d}_1})$ is the stationary point. □

Moreover, by Eq. (49), we have the following corollary.

**Corollary 3.** *If there is a unique stationary point at $[s_{p+1}, s_p)$, $G'(s_p)G'(s_{p+1}) \leq 0$.*

The derivative at the endpoints can be computed as

$$G'(s_p) = \frac{\beta^2}{2\zeta_p^4}\left[\left(\frac{d_2}{\beta} - (p-1)\right)\zeta_p^2 - \sum_{i=p}^{\hat{d}^*} \zeta_i^2\right] = \frac{\beta^2}{2\zeta_p^4}\left[\left(\frac{d_2}{\beta} - p\right)\zeta_p^2 - \sum_{i=p+1}^{\hat{d}^*} \zeta_i^2\right]. \tag{50}$$

Furthermore, once $G'(s_p)$ is non-negative at some endpoint $s_p$, $G'(s) > 0$ holds over $(s_p, \infty)$.

**Lemma 4.** *Let $p, t$ be such that $p \leq d_2/\beta$ and $t > s_p$. Then, $t^2 G'(t) > s_p^2 G'(s_p)$.*

*Proof.* Let $t \in (s_{q+1}, s_q]$ such that $q + 1 \leq p$. We thus have $t > s_{q+1} \geq \cdots \geq s_p$. Because $\frac{d_2}{\beta} - (p-1) > 0$, we have

$$G'(t) = \frac{1}{2t^2}\left[(d_2 - \beta q)t - \sum_{i=q+1}^{\hat{d}^*} \zeta_i^2\right] = \frac{1}{2t^2}\left[(d_2 - \beta(p-1))t - \sum_{i=p}^{q} \beta t - \sum_{i=p}^{\hat{d}^*} \zeta_i^2\right] \tag{51}$$

$$> \frac{1}{2t^2}\left[(d_2 - \beta(p-1))t - \sum_{i=p}^{\hat{d}^*} \zeta_i^2\right] = \frac{s_p^2}{t^2}G'(s_p). \tag{52}$$

□

**Proposition 6.** *$G(s)$ has at most one stationary point over $(0, \infty)$ when $(d_2 - \beta \hat{d}_1) \neq 0$.*

*Proof.* We prove this by contradiction. Let $s_a < s_b$ are two stationary points. Lemma 3 implies these two stationary points are located in two intervals $[s_{p_a+1}, s_{p_a})$ and $[s_{p_b+1}, s_{p_b})$ with $p_a > p_b$. By Corollary 3 there exists $s_l \in \{s_{p_a+1}, s_{p_a}\}$ such that $G'(s_l) \geq 0$, and $s_r \in \{s_{p_b+1}, s_{p_b}\}$ such that $G'(s_r) \leq 0$. Noticing that $s_{p_a+1} < s_{p_a} \leq s_{p_b+1} < s_{p_b}$, we conclude that $s_l \leq s_r$. If $s_l < s_r$, it contradicts to Lemma 4. If $s_l = s_r = s_{p_a}$, then there is no stationary points in the interval $[s_{p_a+1}, s_{p_a})$, which contradicts to the assumption. $\square$

Now, we check whether $0$ and $\infty$ are minima. For $s_+ = s_1 + \sum_{i=1}^{\hat{d}^*} \zeta_i^2 \in [s_1, +\infty)$, we have

$$G'(s_+) = \frac{1}{2s_+^2} \left[ d_2 s_+ - \sum_{i=1}^{\hat{d}^*} \zeta_i^2 \right] = \frac{1}{2s_+^2} \left[ d_2 s_1 + (d_2 - 1) \sum_{i=1}^{\hat{d}^*} \zeta_i^2 \right] > 0, \tag{53}$$

which implies that the $\infty$ is not a minimum.

The behavior of $G'(s)$ in $(0, s_{\hat{d}_1})$ is more complicated.

$$G'(s) = \frac{1}{2s^2} \left[ \left( \frac{d_2}{\beta} - \hat{d}_1 \right) s - \sum_{i=\hat{d}_1+1}^{\hat{d}^*} \zeta_i^2 \right], \tag{54}$$

the sign of which is different for the following three different cases:

1. $\hat{d}_1 = \hat{d}^*$ and $d_2/\beta - \hat{d}_1 > 0$;
2. $\hat{d}_1 = \hat{d}^*$ and $d_2/\beta - \hat{d}_1 = 0$;
3. $\hat{d}_1 < \hat{d}^*$ or $d_2/\beta - \hat{d}_1 < 0$.

Case 1: $\hat{d}_1 = \hat{d}^*$ and $d_2/\beta - \hat{d}_1 > 0$. When $\hat{d}_1 = \hat{d}^*$ and $d_2/\beta - \hat{d} > 0$, $\sum_{i=\hat{d}+1}^{\hat{d}^*} \zeta_i^2 = 0$ and thus $G'(s) > 0$ in $(0, s_{\hat{d}_1}]$. By Lemma 4, $G'(s) > 0$ for $s > s_{\hat{d}_1}$. Therefore, $G'(s) > 0$ for $s \in (0, +\infty)$. Then, there is no global minima for $s \in (0, +\infty)$. The loss function $L_{\text{VAE}}^{\text{LDV}}$ is ill-posed. Even though $s^*$ is converged to $0$, the model is deterministic.

Case 2: $\hat{d}_1 = \hat{d}^*$ and $d_2/\beta - \hat{d}_1 = 0$. When $\hat{d}_1 = \hat{d}^*$ and $d_2/\beta - \hat{d} = 0$, we have $\sum_{i=\hat{d}+1}^{\hat{d}^*} \zeta_i^2 = 0$ and thus $G'(s_{\hat{d}}) = 0$ for all $s \in (0, s_{\hat{d}_1})$. $G'(s_{\hat{d}}) = 0$ also holds by the continuity of $G'(s)$. For any $s > s_{\hat{d}_1}$, $G'(s) > 0$ by Lemma 4. Then the global minima for of $G(s)$ is the entire set of $(0, s_{\hat{d}_1}]$. In such a case, no posterior collapse happens.

Case 3: $\hat{d}_1 < \hat{d}^*$ or $d_2/\beta - \hat{d}_1 < 0$.[4] The following proposition shows that the global minimum of $G'(s)$ is unique.

**Proposition 7.** *When $\hat{d}_1 < \hat{d}^*$ or $d_2/\beta - \hat{d}_1 < 0$, $G(s)$ has a unique global minimum, which is the unique stationary point.*

*Proof.* We first prove the existence. Let $s_- = \min \left\{ s_{\hat{d}_1}, \sum_{i=\hat{d}_1+1}^{d^*} \zeta_i^2 / \left( \frac{d_2}{\beta} - \hat{d}_1 \right) \right\}$. Then,

$$G'(s) = \frac{1}{2s^2} \left[ \left( \frac{d_2}{\beta} - \hat{d}_1 \right) s - \sum_{i=\hat{d}_1+1}^{d^*} \zeta_i^2 \right] < 0 \tag{55}$$

holds in $(0, s_-)$. Recall that $G'(s_+) > 0$ in Eq. (53). Then,

$$G'(s) = \frac{1}{2s^2} \left[ \frac{d_2}{\beta} s - \sum_{i=\hat{d}_1+1}^{d^*} \zeta_i^2 \right] > \frac{1}{2s^2} \left[ \frac{d_2}{\beta} s_+ - \sum_{i=\hat{d}_1+1}^{d^*} \zeta_i^2 \right] > 0 \tag{56}$$

---

[4]The case where $\hat{d}_1 < \hat{d}^*$ and $\beta = 1$ is the case discussed in Lucas et al. (2019) and a variant of the case considered in (Nakajima et al., 2015)

holds in $(s_+, \infty)$. Meanwhile, the continuous function $G$ has minima in the closed interval $[s_-, s_+]$. Then, there exists global minima of $G(s)$ in $(0, \infty)$.

We prove the uniqueness by contradiction. Suppose there are two different global minima such that $s_a^* < s_b^*$. On the one hand, if there are two $p_a < p_b$ such that $s_a^* \in [s_{p_a+1}, s_{p_a})$ and $s_b^* \in [s_{p_b+1}, s_{p_b})$, we have $G'(s_{p_a+1}) > 0$. At the same time, $s_b^*$ is the global minimum implies $G'(s_{p_b+1}) \le 0$, which contradicts to the Lemma 4. On the other hand, if there is a unique $p_a$ such that $s_a^*, s_b^* \in [s_{p_a+1}, s_{p_a})$, $G'(s_a^*) = G'(s_b^*) = 0$, that is $(d_2/\beta - \hat{p}_a)s_a^* = (d_2/\beta - \hat{p}_a)s_b^*$. This implies $d_2/\beta - \hat{p}_a = 0$. Therefore, $G'(s_b^*) = -\frac{1}{2s_b^{*2}} \sum_{i=\hat{p}_a+1}^{d^*} \zeta_i^2 = 0$, which contradicts the proposition assumption.

By Proposition 6, the global minimum of differentiable function $G$ is also the stationary point. $\square$

Now, we are ready to find the optimal $s^*$.

**Theorem 4.** *When $\hat{d}_1 < \hat{d}^*$ or $d_2/\beta - \hat{d}_1 < 0$,*

- *The optimal decoding variance is $\eta_{\text{dec}}^{*2} = \frac{\sum_{i=\hat{d}_1+1}^{\hat{d}^*} \zeta_i^2}{d_2 - \beta \hat{d}_1} \in (0, s_{\hat{d}_1})$ if and only if*

$$\beta < \frac{d_2 \zeta_{\hat{d}_1}^2}{d_1 \zeta_{\hat{d}_1}^2 + \sum_{i=\hat{d}_1+1}^{\hat{d}^*} \zeta_i^2}. \tag{57}$$

- *The optimal decoding variance is $\eta_{\text{dec}}^{*2} = \frac{\sum_{i=p+1}^{\hat{d}^*} \zeta_i^2}{d_2 - \beta p} \in (s_{p+1}, s_p)$, for $1 \le p < \hat{d}_1$ if and only if*

$$\frac{d_2 \zeta_{p+1}^2}{\sum_{i=p+2}^{\hat{d}^*} \zeta_i^2 + (p+1)\zeta_{p+1}^2} \le \beta < \frac{d_2 \zeta_p^2}{\sum_{i=p+1}^{\hat{d}^*} \zeta_i^2 + p\zeta_p^2}. \tag{58}$$

- *The optimal decoding variance is $\eta_{\text{dec}}^{*2} = \frac{1}{d_2} \sum_{i=1}^{\hat{d}^*} \zeta_i^2 \in (s_1, \infty)$ if and only if*

$$\beta \ge \frac{d_2 \zeta_1^2}{\sum_{i=1}^{\hat{d}^*} \zeta_i^2}. \tag{59}$$

*Proof.* To ensure $s^* \in (0, s_{\hat{d}_1})$, then the condition for $\beta$ can be derived by letting $G'(s_{\hat{d}_1}) > 0$, that is

$$\left(\frac{d_2}{\beta} - \hat{d}_1\right)\zeta_{\hat{d}_1}^2 - \sum_{i=\hat{d}_1+1}^{\hat{d}^*} \zeta_i^2 > 0. \tag{60}$$

The optimal $s$ is solved by $G'(s) = 0$, that is

$$(d_2 - \beta \hat{d}_1)s - \sum_{i=\hat{d}_1+1}^{\hat{d}^*} \zeta_i^2 = 0. \tag{61}$$

To ensure $s^* \in [s_{p+1}, s_p)$, then the condition for $\beta$ can be derived by letting $G'(s_p) > 0 \ge G'(s_{p+1})$, that is

$$\left(\frac{d_2}{\beta} - p\right)\zeta_p^2 - \sum_{i=p+1}^{d^*} \zeta_i^2 > 0 \ge \left(\frac{d_2}{\beta} - (p+1)\right)\zeta_{p+1}^2 - \sum_{i=p+2}^{d^*} \zeta_i^2. \tag{62}$$

Then the optimal $s^*$ is solved by $G'(s) = 0$, that is

$$(d_2 - \beta p)s - \sum_{i=p+1}^{\hat{d}^*} \zeta_i^2 = 0. \tag{63}$$

To ensure $s^* \in [s_1, +\infty)$, that is, $G'(s_1) \le 0$. The condition for $\beta$ can be derived by solving

$$\frac{d_2}{\beta}\zeta_1^2 - \sum_{i=1}^{d^*} \zeta_i^2 \le 0. \tag{64}$$

Table 1: The effect of $\beta$ for posterior collapse with learnable decoding variance

| Dimension | $\beta$ Range | Posterior Collapse | $\eta^{*2}_{\text{dec}}$ |
|---|---|---|---|
| $\hat{d}_1 = \hat{d}^*$ | $(0, d_2/\hat{d}_1)$ | NA | NA |
| $\hat{d}_1 = \hat{d}^*$ | $\{d_2/\hat{d}_1\}$ | No collapse or $\zeta^2_{\hat{d}_1}$ only | $(0, s_{\hat{d}_1}]$ |
| $\hat{d}_1 < \hat{d}^*$ | $\left(0, \dfrac{d_2\zeta^2_{\hat{d}_1}}{d_1\zeta^2_{\hat{d}_1}+\sum_{i=\hat{d}_1+1}^{\hat{d}^*}\zeta^2_i}\right)$ | No collapse | $\dfrac{\sum_{i=\hat{d}_1+1}^{\hat{d}^*}\zeta^2_i}{d_2-\beta\hat{d}_1}$ |
| $\hat{d}_1 \le \hat{d}^*$ | $\left[\dfrac{d_2\zeta^2_{p+1}}{\sum_{i=p+2}^{\hat{d}_1}\zeta^2_i+(p+1)\zeta^2_{p+1}}, \dfrac{d_2\zeta^2_p}{\sum_{i=p+1}^{\hat{d}_1}\zeta^2_i+p\zeta^2_p}\right)$ | Partial collapse except the first $p$ modes, $1 \le p < \hat{d}_1$ | $\dfrac{\sum_{i=p+1}^{\hat{d}^*}\zeta^2_i}{d_2-\beta p}$ |
| $\hat{d}_1 \le \hat{d}^*$ | $\left[\dfrac{d_2\zeta^2_1}{\sum_{i=1}^{\hat{d}^*}\zeta^2_i}, +\infty\right)$ | Complete collapse | $\dfrac{1}{d_2}\sum_{i=1}^{\hat{d}^*}\zeta^2_i$ |

Optimal $s^*$ can be found by solving $G'(s) = 0$, that is

$$d_2 s^* - \sum_{i=1}^{\hat{d}^*}\zeta^2_i = 0. \tag{65}$$

$\square$

**Remark.** *By Theorem 2, the posterior collapse for an eigenvalue $\zeta^2_i$ happens when $\beta\eta^2_{\text{dec}} \ge \zeta^2_i$, which is equivalent to $s \ge s_p$ for $1 \le p \le \hat{d}_1$. Therefore, different types of posterior collapse are related to the following conditions of $s^*$*

- *No collapse: $s \in (0, s_{\hat{d}_1})$;*
- *Partial collapse: $s \in [s_{p+1}, s_p)$ for $1 \le p \le \hat{d}_1$;*
- *Complete collapse: $s \in [s_1, \infty)$.*

*Notably, our result shows that the linear VAE with learnable decoding variance does not suffice to lead to no collapse. For an arbitrary choice of $\zeta^2_i$, the condition for no posterior collapse is $\beta \in (0, d_2/\hat{d}^*)$. When $d_2 = d_0 = d^*$, $d_1 = \hat{d}_1$, and $\beta = 1$, the third case reduces to the result of Lucas et al. (2019). However, posterior collapse also happens in this case. For example, when $\zeta^2_1 = ... = \zeta^2_{\hat{d}^*}$, the condition for the complete collapse of the model in Lucas et al. (2019) is $[1, \infty)$, which covers the current choice of $\beta = 1$.*

To summarize, Table 1 concludes five situations for posterior collapse under various conditions.

# E Proofs

## E.1 Proof of Proposition 1

*Proof.* Minimizing $L_{\mathrm{VAE}}(U,W)$ in Eq. (5) is equivalent to the following minimization problem

$$\min_{U,W} \mathbb{E}_x \|UW^\top x - y\|^2 + \mathrm{Tr}(U\Sigma U^\top) + \beta \frac{\eta_{\mathrm{dec}}^2}{\eta_{\mathrm{enc}}^2} \mathrm{Tr}(W^\top AW). \tag{66}$$

It is assumed that $\tilde{x} := \Phi^{-\frac{1}{2}} P_A^\top x$, and $x := P_A \Phi^{\frac{1}{2}} \tilde{x}$. By defining $V := \Phi^{\frac{1}{2}} P_A^\top W$, we obtain

$$\mathbb{E}_x \|UW^\top x - y\|^2 + \mathrm{Tr}(U\Sigma U^\top) + \beta \frac{\eta_{\mathrm{dec}}^2}{\eta_{\mathrm{enc}}^2} \mathrm{Tr}(W^\top AW) \tag{67}$$

$$= \mathbb{E}_x \|UW^\top P_A \Phi^{\frac{1}{2}} \tilde{x} - y\|^2 + \mathrm{Tr}(U\Sigma U^\top) + \beta \frac{\eta_{\mathrm{dec}}^2}{\eta_{\mathrm{enc}}^2} \mathrm{Tr}(W^\top P_A \Phi P_A^\top W) \tag{68}$$

$$= \mathbb{E}_x \|UV\tilde{x} - y\|^2 + \mathrm{Tr}(U\Sigma U^\top) + \beta \frac{\eta_{\mathrm{dec}}^2}{\eta_{\mathrm{enc}}^2} \|V\|_F^2 \tag{69}$$

$$= \mathrm{Tr}(U^\top UV^\top V - 2U^\top ZV) + \mathbb{E}_{\tilde{x}}[y^\top y] + \mathrm{Tr}(U\Sigma U^\top) + \beta \frac{\eta_{\mathrm{dec}}^2}{\eta_{\mathrm{enc}}^2} \mathrm{Tr}(V^\top V) \tag{70}$$

$$= \|UV^\top - Z\|_F^2 + \mathrm{Tr}(U\Sigma U^\top) + \beta \frac{\eta_{\mathrm{dec}}^2}{\eta_{\mathrm{enc}}^2} \|V\|_F^2 - \|Z\|_F^2 + \mathbb{E}_{\tilde{x}}[y^\top y] \tag{71}$$

$$= \|UV^\top - Z\|_F^2 + \mathrm{Tr}(U\Sigma U^\top) + \beta \frac{\eta_{\mathrm{dec}}^2}{\eta_{\mathrm{enc}}^2} \|V\|_F^2, \tag{72}$$

where we have used the relation $\mathbb{E}[\tilde{x}\tilde{x}^T] = I$ and $\|Z\|_F^2 = \mathbb{E}_{\tilde{x}}[y^\top y]$. Thus, the desired $(U,V)$ can be obtained from minimizing $L(U,V) = \|U^\top V - Z\|_F^2 + \mathrm{Tr}(U\Sigma U^\top) + \beta \frac{\eta_{\mathrm{dec}}^2}{\eta_{\mathrm{enc}}^2} \|V\|_F^2$. This finishes the proof. $\square$

## E.2 Proof of Proposition 2

*Proof.* One of the necessary conditions for the global minimum is the zero gradient of $L(U,V)$. We then find the global minimum under the zero gradient condition. Consider

$$\frac{1}{2} \frac{\partial L(U,V)}{\partial V} = VU^\top U - Z^\top U + \beta \frac{\eta_{\mathrm{dec}}^2}{\eta_{\mathrm{enc}}^2} V = 0, \tag{73}$$

which implies

$$V = Z^\top U \left[ \beta \frac{\eta_{\mathrm{dec}}^2}{\eta_{\mathrm{enc}}^2} I + U^\top U \right]^{-1}. \tag{74}$$

Plugging Eq. (74) into the objective in Eq. (6), we have

$$L(U,V) = \mathrm{Tr}\left[ \left( U^\top U + \beta \frac{\eta_{\mathrm{dec}}^2}{\eta_{\mathrm{enc}}^2} I \right) V^\top V - 2U^\top ZV \right] + \mathrm{Tr}(U\Sigma U^\top) + \|Z\|_F^2 \tag{75}$$

$$= \mathrm{Tr}(U\Sigma U^\top) - \mathrm{Tr}(U^\top ZV) + \|Z\|_F^2 \tag{76}$$

$$= \mathrm{Tr}(U\Sigma U^\top) - \underbrace{\mathrm{Tr}\left[ U^\top ZZ^\top U \left( \beta \frac{\eta_{\mathrm{dec}}^2}{\eta_{\mathrm{enc}}^2} I + U^\top U \right)^{-1} \right]}_{:= J} + \|Z\|_F^2. \tag{77}$$

Consider the SVD of matrix $U = Q\Lambda P$ where $Q \in \mathbb{R}^{d_2 \times d_2}$ and $P \in \mathbb{R}^{d_1 \times d_1}$ are orthogonal matrices, $\Lambda \in \mathbb{R}^{d_2 \times d_1}$ is the rectangular diagonal matrix. Meanwhile, consider

$$\left( \beta \frac{\eta_{\mathrm{dec}}^2}{\eta_{\mathrm{enc}}^2} I + P^\top \Lambda^\top \Lambda P \right)^{-1} = P^\top \left( \beta \frac{\eta_{\mathrm{dec}}^2}{\eta_{\mathrm{enc}}^2} I + \Lambda^\top \Lambda \right)^{-1} P. \tag{78}$$

Let diagonal matrix $\Gamma = \beta \frac{\eta_{\text{dec}}^2}{\eta_{\text{enc}}^2} I + \Lambda^\top \Lambda$. Recall the SVD of $Z = F \Sigma_Z G$, then the Eq. (77) is rewritten as

$$J = \text{Tr}\left[\Lambda^\top \Lambda \Sigma\right] - \text{Tr}\left[(Q \Lambda \Gamma^{-1} \Lambda^\top Q^\top)(F \Sigma_Z \Sigma_Z^\top F^\top)\right]. \tag{79}$$

We note that $\Lambda \Gamma \Lambda^\top$ and $\Sigma_Z \Sigma_Z^\top$ are square diagonal matrices in $\mathbb{R}^{d_2 \times d_2}$. Since $\Sigma_Z \in \mathbb{R}^{d_2 \times d_0}$ and there are only $\min(d_0, d_2)$ non-zero values, i.e., $\zeta_i, i = 1, ..., \min(d_0, d_2)$. We denote $\zeta_i = 0$ for $\min(d_0, d_2) < i \le d_1$ if $d_1 > \min(d_0, d_2)$ for convenience. By von Neumann's Trace Inequality (Von Neumann, 1962), the trace of the product of two real symmetric matrices is upper bounded by the sum of the product of their decreasing eigenvalues, specifically,

$$\text{Tr}\left[(Q \Lambda \Gamma^{-1} \Lambda^\top Q^\top)(F \Sigma_Z \Sigma_Z^\top F^\top)\right] \le \text{Tr}\left[\Lambda \Gamma^{-1} \Lambda^\top \Sigma_Z \Sigma_Z^\top\right]. \tag{80}$$

The equality holds if and only if $Q = F$. Then the lower bound of $J$ is achieved when optimal $Q^* = F$.

$$J \ge \text{Tr}\left[\Lambda^\top \Lambda \Sigma\right] - \text{Tr}\left[\Lambda \Gamma^{-1} \Lambda^\top \Sigma_Z \Sigma_Z^\top\right] = \underbrace{\sum_{i=1}^{d_1} \sigma_i^2 \lambda_i^2 - \frac{\zeta_i^2 \eta_{\text{enc}}^2 \lambda_i^2}{\beta \eta_{\text{dec}}^2 + \eta_{\text{enc}}^2 \lambda_i^2}}_{:= J_{Q^*}}. \tag{81}$$

$J_{Q^*}$ can be further minimized over all $\lambda_i$. The optimal $\lambda_i^*$ can be determined by setting the corresponding gradients to zero. Consider $t_i = \lambda_i^2 \ge 0$,

$$\frac{\partial J_{Q^*}}{\partial t_i} = \frac{\partial}{\partial t_i}\left[\sigma_i^2 t_i - \frac{\zeta_i^2 \eta_{\text{enc}}^2 t_i}{\beta \eta_{\text{dec}}^2 + \eta_{\text{enc}}^2 t_i}\right] \tag{82}$$

$$= \sigma_i^2 - \frac{\zeta_i^2 \eta_{\text{enc}}^2 (\beta \eta_{\text{dec}}^2 + \eta_{\text{enc}}^2 t_i) - \eta_{\text{enc}}^2 \zeta_i^2 \eta_{\text{enc}}^2 t_i}{(\beta \eta_{\text{dec}}^2 + \eta_{\text{enc}}^2 t_i)^2} \tag{83}$$

$$= \sigma_i^2 - \frac{\zeta_i^2 \eta_{\text{enc}}^2 \beta \eta_{\text{dec}}^2}{(\beta \eta_{\text{dec}}^2 + \eta_{\text{enc}}^2 t_i)^2} \tag{84}$$

$$= \frac{\sigma_i^2 (\beta \eta_{\text{dec}}^2 + \eta_{\text{enc}}^2 t_i)^2 - \zeta_i^2 \eta_{\text{enc}}^2 \beta \eta_{\text{dec}}^2}{(\beta \eta_{\text{dec}}^2 + \eta_{\text{enc}}^2 t_i)^2} = 0. \tag{85}$$

Two solutions of the Eq. (85) are

$$t_i^{(1)} = \frac{\sqrt{\beta} \eta_{\text{dec}}}{\sigma_i \eta_{\text{enc}}}\left(\zeta_i - \frac{\sqrt{\beta} \sigma_i \eta_{\text{dec}}}{\eta_{\text{enc}}}\right), \tag{86}$$

$$t_i^{(2)} = \frac{\sqrt{\beta} \eta_{\text{dec}}}{\sigma_i \eta_{\text{enc}}}\left(-\zeta_i - \frac{\sqrt{\beta} \sigma_i \eta_{\text{dec}}}{\eta_{\text{enc}}}\right). \tag{87}$$

We see that $t_i \ge 0 > t_i^{(2)}$, then the monotonicity of $J_Q^*$ with respect to $t_i$ over $(0, +\infty)$ only depends on $t_i^{(1)}$. Here are two situations: (1) $t_i^{(1)} \le 0$: $\frac{\partial J_{Q^*}}{\partial t_i} \ge 0$, then $J_Q^*$ increases monotonically with $t_i$. Then the optimal $t_i^* = 0$. (1) $t_i^{(1)} > 0$: $\frac{\partial J_{Q^*}}{\partial t_i} > 0$ when $t_i > t_i^{(1)}$ and $\frac{\partial J_{Q^*}}{\partial t_i} < 0$ when $t_i < t_i^{(1)}$. Then the optimal $t_i^* = t_i^{(1)}$. Therefore, optimal $\lambda_i^*$ is summarized by the two situations above with $\lambda_i^{*2} = t_i^*$

$$\lambda_i^* = \sqrt{\max\left(0, \frac{\sqrt{\beta} \eta_{\text{dec}}}{\sigma_i \eta_{\text{enc}}}\left(\zeta_i - \frac{\sqrt{\beta} \sigma_i \eta_{\text{dec}}}{\eta_{\text{enc}}}\right)\right)}, i = 1, ..., d_1. \tag{88}$$

As a result, $U = Q^* \Lambda^* P$ where $P$ is an arbitrary orthogonal matrix in $\mathbb{R}^{d_1 \times d_1}$. The optimal $V^*$ can also be determined by Eq. (74)

$$V^* = \bar{G} \Theta P, \tag{89}$$

where $\bar{G} = [g_1, ..., g_{d_1}]$, $\Theta = \text{diag}(\theta_1, ..., \theta_{d_1})$ where $\theta_i = \sqrt{\max\left(0, \frac{\sigma_i \eta_{\text{enc}}}{\sqrt{\beta} \eta_{\text{dec}}}\left(\zeta_i - \frac{\sqrt{\beta} \sigma_i \eta_{\text{dec}}}{\eta_{\text{enc}}}\right)\right)}$. $\square$

### E.3 Proof of Corollary 1

*Proof.* The minimum value can be obtained by plugging in the optimal

$$\lambda_i^* = \sqrt{\max\left(0, \frac{\sqrt{\beta}\eta_{\text{dec}}}{\sigma_i \eta_{\text{enc}}}\left(\zeta_i - \frac{\sqrt{\beta}\sigma_i \eta_{\text{dec}}}{\eta_{\text{enc}}}\right)\right)},\tag{90}$$

into the lower bound of $L(u, v)$, i.e.,

$$L(U, V) \geq \min_{U,V} L(U, V) = \|Z\|_F^2 + J_{Q^*} = \sum_{i=1}^{d_1} \zeta_i^2 + \sigma_i^2 \lambda_i^{*2} - \frac{\zeta_i^2 \eta_{\text{enc}}^2 \lambda_i^{*2}}{\beta \eta_{\text{dec}}^2 + \eta_{\text{enc}}^2 \lambda_i^{*2}}.\tag{91}$$

$\square$

### E.4 Proof of Proposition 3

*Proof.* The optimal $\sigma_i$ can be determined by

$$\sigma_i^* = \arg\min_{\sigma>0} l_i(\sigma) = \arg\min_{\sigma>0} \zeta_i^2 - \left(\zeta_i - \frac{\sqrt{\beta}\sigma_i \eta_{\text{dec}}}{\eta_{\text{enc}}}\right)^2 \mathbb{1}_{\zeta_i > \frac{\sqrt{\beta}\sigma_i \eta_{\text{dec}}}{\eta_{\text{enc}}}} + \beta \eta_{\text{dec}}^2\left(\frac{\sigma_i^2}{\eta_{\text{enc}}^2} - 1 - \log\frac{\sigma_i^2}{\eta_{\text{enc}}^2}\right).\tag{92}$$

The gradient of $l_i(\sigma)$ reads,

$$l_i'(\sigma) = \mathbb{1}_{\zeta_i > \frac{\sqrt{\beta}\eta_{\text{dec}}}{\eta_{\text{enc}}}\sigma}\left(\zeta_i - \frac{\sqrt{\beta}\eta_{\text{dec}}}{\eta_{\text{enc}}}\sigma\right)\frac{\sqrt{\beta}\eta_{\text{dec}}}{\eta_{\text{enc}}} + \frac{\beta \eta_{\text{dec}}^2}{\eta_{\text{enc}}^2}\left(\sigma - \frac{\eta_{\text{enc}}^2}{\sigma}\right)\tag{93}$$

Since $l'(\sigma)$ is a increasing function, $l'(\sigma_-) < 0$ when $\sigma_- = \frac{1}{2}\min\left(\frac{\zeta_i \eta_{\text{enc}}}{\sqrt{\beta}\eta_{\text{dec}}}, \frac{\sqrt{\beta}\eta_{\text{enc}}\eta_{\text{dec}}}{\zeta_i}\right)$, and $l'(\sigma_+) > 0$ when $\sigma_- = 2\max\left(\frac{\zeta_i \eta_{\text{enc}}}{\sqrt{\beta}\eta_{\text{dec}}}, 2\eta_{\text{enc}}\right)$. Then the minimal value of $l(\cdot)$ is determined when $l'(\sigma) = 0$, that is,

$$\mathbb{1}_{\sigma < \zeta_i \frac{\eta_{\text{enc}}}{\sqrt{\beta}\eta_{\text{dec}}}}\frac{\sqrt{\beta}\eta_{\text{dec}}}{\eta_{\text{enc}}}\zeta_i + \left[1 - \mathbb{1}_{\sigma < \zeta_i \frac{\eta_{\text{enc}}}{\sqrt{\beta}\eta_{\text{dec}}}}\right]\frac{\beta \eta_{\text{dec}}^2}{\eta_{\text{enc}}^2}\sigma = \frac{\beta \eta_{\text{dec}}^2}{\sigma}.\tag{94}$$

The LHS of Equation (94) is a non-decreasing function while the RHS is decreasing function. Then we claim there is a unique solution $\sigma^*$ of Equation (94). The solution breaks down into two situations

**Case 1:** $\sigma^* < \zeta_i \frac{\eta_{\text{enc}}}{\sqrt{\beta}\eta_{\text{dec}}}$

In this case, we have

$$\frac{\sqrt{\beta}\eta_{\text{dec}}\zeta_i}{\eta_{\text{enc}}} = \frac{\beta \eta_{\text{dec}}^2}{\sigma}.\tag{95}$$

Then

$$\sigma^* = \frac{\sqrt{\beta}\eta_{\text{dec}}\eta_{\text{enc}}}{\zeta_i}.\tag{96}$$

This solution holds if and only if the following condition holds

$$\frac{\sqrt{\beta}\eta_{\text{dec}}\eta_{\text{enc}}}{\zeta_i} < \zeta_i \frac{\eta_{\text{enc}}}{\sqrt{\beta}\eta_{\text{dec}}} \Leftrightarrow \beta \eta_{dec}^2 < \zeta_i^2.\tag{97}$$

**Case 2:** $\sigma^* \geq \zeta_i \frac{\eta_{\text{enc}}}{\sqrt{\beta}\eta_{\text{dec}}}$

In this case, we have

$$\frac{\beta \eta_{\text{dec}}^2}{\eta_{\text{enc}}^2}\sigma = \frac{\beta \eta_{\text{dec}}^2}{\sigma}.\tag{98}$$

Then

$$\sigma^* = \eta_{\text{enc}}.\tag{99}$$

This solution holds when

$$\eta_{\text{enc}} > \zeta_i \frac{\eta_{\text{enc}}}{\sqrt{\beta}\eta_{\text{dec}}} \Leftrightarrow \beta \eta_{dec}^2 \geq \zeta_i^2.\tag{100}$$

It is easy to check that the two cases above cover all solutions. $\square$

### E.5 Proof of Theorem 3

*Proof.* The first-order derivatives of $L$ are

$$\frac{\partial L(U,V)}{\partial U} = 2\left(UV^\top V - ZV + U\Sigma\right) \tag{101}$$

$$\frac{\partial L(U,V)}{\partial V} = 2\left(VU^\top U - Z^\top U + \beta\frac{\eta_{\text{dec}}^2}{\eta_{\text{enc}}^2}V\right) \tag{102}$$

Then the second-order derivatives of $L$ are

$$\frac{\partial^2 L(U,V)}{\partial U_{rs}\partial U_{pq}} = 2\delta_{pr}\left(\sum_{k=1}^{d_0} V_{ks}V_{kq} + \delta_{qs}\sigma_q^2\right) \tag{103}$$

$$\frac{\partial^2 L(U,V)}{\partial V_{rs}\partial U_{pq}} = 2\left[U_{ps}V_{rq} + \left(\sum_{j=1}^{d_1} U_{pj}V_{rj} - Z_{pr}\right)\delta_{qs}\right] \tag{104}$$

$$\frac{\partial^2 L(U,V)}{\partial U_{rs}\partial V_{pq}} = 2\left[V_{ps}U_{rq} + \left(\sum_{j=1}^{d_1} U_{rj}V_{pj} - Z_{rp}\right)\delta_{qs}\right] \tag{105}$$

$$\frac{\partial^2 L(U,V)}{\partial V_{rs}\partial V_{pq}} = 2\delta_{pr}\left(\sum_{i=1}^{d_2} U_{is}U_{iq} + \beta\frac{\eta_{\text{dec}}^2}{\eta_{\text{enc}}^2}\delta_{qs}\right). \tag{106}$$

Letting $U = 0$ and $V = 0$

$$\left.\frac{\partial^2 L(U,V)}{\partial U_{rs}\partial U_{pq}}\right|_{U=0,V=0} = 2\delta_{pr}\delta_{qs}\sigma_q^2 \tag{107}$$

$$\left.\frac{\partial^2 L(U,V)}{\partial V_{rs}\partial U_{pq}}\right|_{U=0,V=0} = -2Z_{pr}\delta_{qs} \tag{108}$$

$$\left.\frac{\partial^2 L(U,V)}{\partial U_{rs}\partial V_{pq}}\right|_{U=0,V=0} = -2Z_{rp}\delta_{qs} \tag{109}$$

$$\left.\frac{\partial^2 L(U,V)}{\partial V_{rs}\partial V_{pq}}\right|_{U=0,V=0} = 2\beta\frac{\eta_{\text{dec}}^2}{\eta_{\text{enc}}^2}\delta_{pr}\delta_{qs}. \tag{110}$$

Then we consider the quadratic form at $U = 0, V = 0$. Consider $\Delta U$ and $\Delta V$ as the perturbation of $U$ and $V$. Then the quadratic form reads

$$LQ(\Delta U, \Delta V) = \left[\sum_{pqrs}\left.\frac{\partial^2 L(U,V)}{\partial U_{rs}\partial U_{pq}}\right|_{U=0,V=0}\Delta U_{rs}\Delta U_{pq} + \sum_{pqrs}\left.\frac{\partial^2 L(U,V)}{\partial V_{rs}\partial U_{pq}}\right|_{U=0,V=0}\Delta V_{rs}\Delta U_{pq}\right.$$
$$\tag{111}$$

$$\left.+ \sum_{pqrs}\left.\frac{\partial^2 L(U,V)}{\partial U_{rs}\partial V_{pq}}\right|_{U=0,V=0}\Delta U_{rs}\Delta V_{pq} + \sum_{pqrs}\left.\frac{\partial^2 L(U,V)}{\partial V_{rs}\partial V_{pq}}\right|_{U=0,V=0}\Delta V_{rs}\Delta V_{pq}\right]$$
$$\tag{112}$$

$$= 2\sum_{pq}\sigma_q^2\Delta U_{pq}^2 - 2\sum_{pqr}Z_{pr}\Delta U_{pq}\Delta V_{rq} - 2\sum_{pqr}Z_{rp}\Delta U_{rq}\Delta V_{pq} + \sum_{pq}2\beta\frac{\eta_{\text{dec}}^2}{\eta_{\text{enc}}^2}\Delta V_{pq}^2$$
$$\tag{113}$$

$$= 2\left[\text{Tr}(\Delta U\Sigma(\Delta U)^\top) - 2\text{Tr}(Z\Delta V(\Delta U)^\top) + \beta\frac{\eta_{\text{dec}}^2}{\eta_{\text{enc}}^2}\text{Tr}(\Delta V(\Delta V)^\top)\right] \tag{114}$$

It suffices to consider the case $\|X\|_F^2 = 1$. Let $\alpha^2 = \|\Delta U\|_F^2$ and $\|\Delta V\|_F^2 = 1 - \alpha^2$. Then let $u = \Delta U/\alpha$ and $v = \Delta V/\sqrt{1-\alpha^2}$ be the normalized matrix. Plugging in, we obtain

$$LQ(\Delta U, \Delta V) \propto \sigma^2 \|U\|^2 - 2\mathrm{Tr}(ZVU^\top) + \beta \frac{\eta_{\text{dec}}^2}{\eta_{\text{enc}}^2} \|V\|^2 \tag{115}$$

$$= \sigma^2 \alpha^2 - 2\sqrt{\alpha^2(1-\alpha^2)}\mathrm{Tr}(u^\top Zv) + \beta \frac{\eta_{\text{dec}}^2}{\eta_{\text{enc}}^2}(1-\alpha^2), \tag{116}$$

where we assume $\Sigma = \sigma I$, according to the Theorem 2. Apparently, for any fixed $\alpha$, the middle term is minimized if $u$ is the left eigenvector of $Z$ corresponding to the largest singular value of $Z$, and $v$ is the corresponding right eigenvector. This choice gives

$$LQ(\Delta U, \Delta V) \propto \sigma^2 \alpha^2 - 2\sqrt{\alpha^2(1-\alpha^2)}\zeta_{\max} + \beta \frac{\eta_{\text{dec}}^2}{\eta_{\text{enc}}^2}(1-\alpha^2). \tag{117}$$

Minimizing over $\alpha$ shows that

$$\min_\alpha LQ(\Delta U, \Delta V) \propto \sigma^2 + \beta \frac{\eta_{\text{dec}}^2}{\eta_{\text{enc}}^2} - \sqrt{\left(\sigma^2 - \beta \frac{\eta_{\text{dec}}^2}{\eta_{\text{enc}}^2}\right)^2 + 4\zeta_{\max}^2} \tag{118}$$

which is nonnegative if and only if $\zeta_{\max}^2 \geq \sigma^2 \beta \frac{\eta_{\text{dec}}^2}{\eta_{\text{enc}}^2}$. Namely, $\sigma^2 \beta \frac{\eta_{\text{dec}}^2}{\eta_{\text{enc}}^2} - \zeta_{\max}^2 < 0$ implies that the origin is a saddle point. Meanwhile, $\sigma^2 \beta \frac{\eta_{\text{dec}}^2}{\eta_{\text{enc}}^2} - \zeta_{\max}^2 > 0$ implies that the origin is a local minimum. Notice that this condition coincides with the condition that the origin is a global minimum. Therefore, the origin is the global minimum if and only if the Hessian at the origin is PSD. This finishes the proof. $\qquad\square$