# OpenReview forum: "Posterior Collapse of a Linear Latent Variable Model"
_NeurIPS.cc/2022/Conference — NeurIPS 2022 Accept_

### Official Review · Reviewer_pzUk · 2022-06-21

**Rating:** 8
**Confidence:** 3
**Soundness:** 3 good
**Presentation:** 4 excellent
**Contribution:** 3 good

**Summary:**

The paper provides a theoretical analysis of (a simplified) linear VAE in order to improve our understanding of the posterior collapse often observed with similar nonlinear Bayesian models. The main finding is that these models naturally perform a thresholding of singular values, which is at the root of the issue. This can then be linked to the prior on the mean.

**Questions:**

* Line 217 (a): it is argued that a learnable variance does not change the posterior collapse. This is under the assumption that the learnable variance is data independent. Do the comments still hold true if the variance is data dependent (as is the common case) ?
* Line 262-275: It is observed that an optimum may exist where the parameters are equal to zero. Is it correctly understood that this is due to the data being zero mean, i.e. that the mode is for the model to predict the data mean?

**Limitations:**

The most fundamental limitation of the paper is that it studies a linear model in the hope that this may tell us something about nonlinear models. Here we can only hope. The paper is quite clear on this limitation.

A second limitation is that all noise variances are assumed to be data independent, which is quite far from common practice (esp. for the encoder) and is at odds with some of the driving ideas of VAEs. I think the paper could be more clear about this limitation.

**Strengths And Weaknesses:**

### Strengths
* The paper is fairly easy to read, and the authors do a nice job of summarizing their findings along the way to keep the reader engaged.
* The main findings are interesting, and inspirational.
* The empirical results on nonlinear models, while not the focus of the paper, provides evidence in support of the findings from linear models may generalize.

### Weakness
* The paper treats both encoder and decoder as data independent, which is quite far from the usual VAE setting. I understand that using linear models to predict uncertainties would be rather useless, but I still think it would be good if the paper acknowledged the limitation.
* I really miss a greater discussion of the link to (P)PCA in the paper. If I understand correctly, it is these links that are examined in the prior work by Lucas et al., but given that a linear VAE is basically a variant of PCA, I found it rather confusing that these links are not discussed in the paper.
* In line 179, it is stated that one solution to the posterior collapse problem is to pick the variance of the likelihood to be small. First, this doesn't really work for non-Gaussian likelihoods (which are the most common), and second it seems like a poor solution to effectively have a delta-likelihood in a probabilistic model (then there is little point in being probabilistic in the first place).

### Minor comments
* Line 105 says that A is full rank, while in 111 it is semidefinite. That's confusing.
* In line 136, it is stated that Sigma is learnable, yet earlier it was not learnable. Again, confusing.
* In lines 299-300 it would be good to use the proper notation $10^{-3}$ rather than 1e-3.
* Late in the paper it is stated that the introduction of beta (as in the beta-VAE) makes little sense from a probabilistic perspective. I think this should be stated when beta is initially introduced.

---

> ### Author Response · Authors · 2022-08-02
> **Author Reply**
>
>
> Thank you very much for appreciating our work. We have added a section in the appendix to study the effect of a data-dependent encoder variance, and we clarify here the questions you raised.
>
> > I really miss a greater discussion of the link to (P)PCA in the paper. If I understand correctly, it is these links that are examined in the prior work by Lucas et al., but given that a linear VAE is basically a variant of PCA, I found it rather confusing that these links are not discussed in the paper.
> - More discussions of pPCA are added in the paper now. See the updated related works section and updated discussion after some of the theorems.
>
> > In line 179, it is stated that one solution to the posterior collapse problem is to pick the variance of the likelihood to be small. First, this doesn't really work for non-Gaussian likelihoods (which are the most common), and second it seems like a poor solution to effectively have a delta-likelihood in a probabilistic model (then there is little point in being probabilistic in the first place).
> - We agree. We only suggested it as a possible “technical” solution but not as a “good” solution. We have made this point clear in the updated draft.
>
> > Line 217 (a): it is argued that a learnable variance does not change the posterior collapse. This is under the assumption that the learnable variance is data independent. Do the comments still hold true if the variance is data dependent (as is the common case) ?
> - Thanks for this suggestion. We have added the theoretical analysis of the case when the encoder variance is dependent on the input data in Section C, and it shows that in the case of a linear dependence on the input data, our results and messages remain unchanged. This also answers the first weakness raised above and the second limitation pointed out below. That being that, we still do not consider the case when the decoder variance is data-dependent, given that it is rare in practice and that we do not think that it would play a significant role in influencing the collapses.
>
>
> > Line 262-275: It is observed that an optimum may exist where the parameters are equal to zero. Is it correctly understood that this is due to the data being zero mean, i.e. that the mode is for the model to predict the data mean?
> - Let us clarify that the main result does not assume that the data is zero-mean (e.g., our result applies to the case when $\mathbb{E}[x] \neq 0$ and/or $\mathbb{E}[y]\neq 0$). Thus, the optimum being zero is not a consequence of the data being zero-mean.

---

### Official Review · Reviewer_R7JS · 2022-07-10

**Rating:** 7
**Confidence:** 4
**Soundness:** 4 excellent
**Presentation:** 3 good
**Contribution:** 1 poor

**Summary:**

This paper analyzes the globally optimal solutions to linear latent variable models. The authors show that these solutions exhibit low-rank structure that is equivalent to so-called posterior collapse.

The paper introduces a "(generalized) variational autoencoder" objective where the input and target spaces need not match. They show (Proposition 1) that solutions of the VAE objective with a fixed approximate posterior variance are aligned with solutions of a related matrix factorization problem (up to an [assumed] full-rank matrix multiplication). The optimal solution of this matrix factorization problem is computed and the (possibly) low-rank structure is characterized.

The results are then generalized (Section 4.3) to the setting where the variance of the approximate posterior is learned (and is assumed to be data-independent, which is optimal in the linear VAE setting [Lucas et al. 2019]). The authors interpret their results in terms of the posterior collapse phenomena that occurs in deep VAEs and empirically investigate their findings on synthetic data and the MNIST dataset.

**Post-rebuttal update:** The authors have made significant changes to their paper following the reviews. With these, they address the vast majority of the issues that I presented below. They also pointed out that my claims on $\beta$ being unnecessary in the linear VAE setting are not technically correct. Following this, I have updated my score to recommend that the paper be accepted. I hope that the authors will continue to improve the presentation of the work.

**Questions:**

I feel that the following is too strong a statement: "Up to now, there has not been any precise identification of neither the nature nor the cause of the posterior collapse problem." See discussion above.

$A$ is positive semi-definite but it is then assumed that $A^{1/2}$ is invertible. This is equivalent to assuming that all eigenvalues of the data covariance matrix are strictly positive, and it is stated in footnote that it is straightforward to extend to the low-rank case. But this is not evident to me from the proofs. For such a claim, ideally, the proofs in the supplementary material would apply to the low-rank case. You could keep the assumption in the main paper if it heavily simplifies notation etc., but I do not quite see how.

Why does notation appear at the end of the problem setting section? There is some duplication here, as much of it has already been defined above.

There is some confusing notation in the statement of Proposition 1. It swaps between $U$ and $W$ with $U^*$ doubly defined as the optimum of both objectives implicitly. Notation is also swapped in the proof. Further, the proof of Proposition 1 claims that these minimization problems are identical. It is have shown that,
$L_{\mathrm{VAE}}(U,W) = L(U, A^{1/2}W)$, which is not quite the claim (particularly in the case where $A$ is not full-rank, then each minimizer of the former may correspond to a manifold of solutions for the latter). The proposition statement compares *the* minimizers, but note that both objectives are non-convex and a unique minimizer has not yet been established. Perhaps a more accurate statement is that any stationary point has these dual forms (under a full-rank $A$).

#### Proof of proposition 2

The proof begins by stating conditions on the global minimum; but this is a non-convex optimization problem and so this condition guarantees only a stationary point. As written, this is not incorrect but is a little confusing. The conditions are established for a stationary point and then solved for the global minimum by finding the stationary parameters with the smallest loss.

I encourage the authors to state the SVD of Z within the proof again (otherwise the definition of F can be a source of temporary confusion).

Please state the trace inequality that you are using exactly (trace of matrix product upper bounded by sum of product of eigenvalues). It is also worth including the derivation of equation 45 --- the proof is difficult to follow without this. Similarly for equation 46 (how did you verify that only the larger solution is related to the minimum?).

#### Proof of proposition 3

It is written that the gradient is an increasing function and so the minimum value is determined when the gradient is zero. You also need to add the condition that $\ell'(\sigma) < 0$ for some $\sigma$ which is then given in closed-form.


Minor comments:

L23: in what sense is this more general? Due to `x` and `y`? Or due to the stochastic reparameterization trick used in VAEs? There is also no marginalization over `y` and it is unclear what `x` and `y` represent here. (This is corrected later, in Section 3).

L34: "in the deep learning context" -> suggests you are doing this. But it is a stretch to claim this when studying linear models only.

L44-45: a strong claim given referenced work. May need further justification.

L245: your posterior collapse condition also depends on the decoder variance. You also write above that the decoder variance can also be controlled to alleviate posterior collapse. And finally, you do not analyze learning the decoder variance but if you included the partition function in the objective I expect that the same analysis of Lucas et al. could be repeated in this setting to the same general effect.

L284: Similar to L245, this is only true when excluding the partition function from the objective (and restricting what we mean by "the effect").

L324: "to [what] extent the result carries"

L326-327: "Moreover, can we design a Bayesian-principled method for avoiding posterior collapse that does not involve hand-tuning β or ηdec?" This is exactly one of the contributions of Lucas et al., in the case of the linear VAE.

**Limitations:**

Limitations are discussed adequately within the paper. However, it is my feeling that the framing of the paper relative to the related work is inaccurate and the discussion of limitations should be updated therein.

**Strengths And Weaknesses:**

## Strengths

The theoretical results are well-connected and the proofs are (mostly) well-written and easy to follow. This is also true of the main paper itself (bar a few minor notational issues).

The authors analyze (a variant of) the beta-VAE objective which is widely used in practice.

## Weaknesses

The objective presented in Equations (2)-(5) is incorrect and does not include the log partition function from the decoder observation model. There is an argument to be made that this matches practical implementations or perhaps that this term can be ignored when the decoder variance is not learned. But as I will discuss further, this is a problematic assumption that has been carefully studied for this setting in prior work.

A limitation of the analysis, relative to Lucas et al., is that only the form of the global optimum is analyzed under this objective and none of the other stationary points are characterized. Moreover, beta plays the role of controlling the rank of the global optimum but in the linear VAE with learned decoder variance there is no need for a beta parameter as all low-rank stationary points become saddle points. I suspect that all other stationary points here are also saddles, but many of the higher rank solutions have larger loss until beta is reduced. Further to this point, the authors write that "one alternative way to fix posterior collapse[s] that have not been suggested in the field is to use a sufficiently small [decoder variance]". But this is exactly the suggestion of Lucas et al. (!) where the decoder variance is treated both as a learned or fixed parameter.

It is also not surprising that the choice of the prior variance scale has no effect on posterior collapse. In the models considered (and in deep VAEs), this variance can be trivially accounted for by scaling the encoder and decoder weights up and down respectively. This can be understood easily via appendix B of Lucas et al., where such a transformation is shown under which the likelihood term is invariant --- this (full-rank) transformation can be chosen to adjust for any choice of prior variance.

And finally, the point could be made that beta VAEs are a common method used in practice and certainly they are worthy of study in their own right. Though, note that Lucas et al. discuss beta VAEs in Chapter 5 and draw a clear connection between the role of beta and of the observation noise in the general ELBO objective with Gaussian likelihood. In particular, that when treating both as fixed hyperparameters the exact same gradient update can be computed in either case.

The remaining contribution of this work that is not supported by prior work is that the authors consider more general models than VAEs (with different output dimensions). However, I do not consider this contribution to be significant enough to overcome the negative points discussed elsewhere in my review.

To my knowledge, the authors have provided novel theoretical results. They generalize past results in the sense that they account for a KL-annealing term and a different output dimensionality. But they are weaker in the ways discussed above. Considered in balance, I feel that these contributions are not significant enough to warrant acceptance.

Finally, the empirical results may not be a primary contribution of this work but they are markedly weaker than similar publications at previous venues. The authors train linear VAEs on MNIST and a synthetic dataset. And two-layer fully-connected encoder/decoder networks are also trained on MNIST. [Note that the variance is not global as in the theoretical analysis but is instead data-dependent, but I agree with the authors that this is a reasonable choice.] For comparison, Lucas et al. 2019 train fully-connected VAEs on MNIST, and convolutioanl VAEs on CelebA. Their empirical evaluation included different objectives, varying encoder architectures, KL annealing schedules, fixed/learned decoder variance, and more. Dai et al. 2020 train on MNIST, Fashion-MNIST, SVHN, CIFAR10, CIFAR100, and CelebA. They used fully-connected, convolutional, and residual architectures and evaluated different objectives and model capacity. Shekhovtsov et al. 2022 (see below) have a similarly thorough empirical evaluation. In summary, while I see no significant issue with the empirical evaluation in this paper I also am unable to recommend it as a strength of the work relative to other papers investigating the same issue.

### Missing discussion of related work

The connection between linear VAEs and matrix factorization are already well-known [1,2,3]. In these references, non-amortized variational inference is used but many of the findings overlap with that of this paper. A suitable discussion and comparison must be included.

Some other work investigating posterior collapse has also been missed in discussion [4,5].

[1] S. Nakajima and M. Sugiyama. Implicit regularization in variational bayesian matrix factorization. ICML, 2010.

[2] S. Nakajima, M. Sugiyama, S. D. Babacan, and R. Tomioka. Global analytic solution of fully observed variational bayesian matrix factorization. Journal of Machine Learning Research, 2013.

[3] S. Nakajima, R. Tomioka, M. Sugiyama, and S. D. Babacan. Condition for perfect dimensionality recovery by variational bayesian PCA. Journal of Machine Learning Research, 2015.

[4] A. Shekhovtsov, D. Schlesinger, and B. Flach. VAE approximation error: ELBO and exponential families. ICLR, 2022.

[5] J.Lücke, D. Forster, and Z. Dai. The evidence lower bound of variational autoencoder converges to a sum of three entropies. ICLR, 2022.

---

> ### Author Response · Authors · 2022-08-02
> **Author Reply Part 1**
>
> We would like to thank you for the detailed criticisms and constructive feedback you gave. We have significantly updated the manuscript to make the generality of our analysis extend beyond that of the previous works. In particular, the following three additions make our results more general (and also more practically relevant) than that of Lucas et al.:
> 1. when the decoder variance is learnable as in Lucas et al. (2019) [Appendix Section D]
> 2. when the data covariance $A$ is not invertible [Updated Proposition 1 and its proof]
> 3. when the encoder variance is dependent on the input, which is common in practice [Appendix Section C]
>
> See the detailed reply below. Please let us know if there is any remaining problem, and we are willing to make further updates in future versions. At the same time, we feel that some of your original criticism may have been overly harsh, and we also point this out along the way.
>
>
> > The objective presented in Equations (2)-(5) is incorrect and does not include the log partition function from the decoder observation model. There is an argument to be made that this matches practical implementations or perhaps that this term can be ignored when the decoder variance is not learned. But as I will discuss further, this is a problematic assumption that has been carefully studied for this setting in prior work.
> - First of all, we point out that Eqs. (2)-(5) are correct. They are just different from those of Lucas et al.. We consider the case then the decoder variance is not learned, and, as you already point out: this matches common practice in the field.
> - That being said, we now added the analysis of the case when the decoder variance is learned in Section D. The updated result is more general than that of Lucas et al. because (1) the effect of having the $\beta$ term is included, and is shown to have a crucial impact on the nature of the problem; (2) our result also applies to the case when $d_2\neq d_0$, when $d_1\geq d_2$ and when $d_1\geq d_0$.
>
> >A limitation of the analysis, relative to Lucas et al., is that only the form of the global optimum is analyzed under this objective and none of the other stationary points are characterized.
> - We didn't consider this case because we found global minima sufficient to explain the experimental results. While it is interesting and not difficult to explore the other stationary points, we feel that it is more a digression rather than an addition to our results.
>
> > Moreover, beta plays the role of controlling the rank of the global optimum but in the linear VAE with learned decoder variance there is no need for a beta parameter as all low-rank stationary points become saddle points.
> - We point out that this statement is (1) mathematically incorrect and (2) is insufficient to understand posterior collapse even if it is correct. First of all, the suggestion that “there is no need for a beta parameter as all low-rank stationary points become saddle points” is incorrect. See Eq. (7)-(8) of Lucas et al.. The global minimum solution becomes low-rank when the smallest k+1 eigenvalues of \Lambda are identical. Therefore, even if the decoder variance is learnable, posterior collapse can still happen in principle: this implies that a learnable decoder variance is not sufficient to remedy or account for the cause of the posterior collapse. Even if the statement is true, it is insufficient to suggest that a decoder variance is related to the cause of the posterior collapse. In essence, this criticism is based on the following logic: if applying A (making decoder variance learnable) can remedy B (posterior collapse), then A causes B. This logic is not scientific. Consider the following example. Suppose we have a noisy radio. The fact that removing the battery can remove the noise does not mean that the battery is the cause of the noise.
> - Therefore, it is both reasonable and important to study the case then the decoder variance is not learned, contrary to the criticism that it is “incorrect” to do so.
>
> > Further to this point, the authors write that "one alternative way to fix posterior collapse[s] that have not been suggested in the field is to use a sufficiently small [decoder variance]". But this is exactly the suggestion of Lucas et al. (!) where the decoder variance is treated both as a learned or fixed parameter.
> - This is a misinterpretation of our result. We mentioned this method only in order to criticize it. For example, we suggested that it does not agree with the Bayesian principle. We have updated the statement to avoid claiming this as a contribution.

---

> ### Author Response · Authors · 2022-08-02
> **Author Reply Part 2**
>
> > It is also not surprising that the choice of the prior variance scale has no effect on posterior collapse. In the models considered (and in deep VAEs), this variance can be trivially accounted for by scaling the encoder and decoder weights up and down respectively. This can be understood easily via appendix B of Lucas et al., where such a transformation is shown under which the likelihood term is invariant --- this (full-rank) transformation can be chosen to adjust for any choice of prior variance.
> - We feel that this is an overinterpretation of Appendix B of Lucas et al. Lucas et al. does not point out or discuss its relevance to collapses.
>
> > And finally, the point could be made that beta VAEs are a common method used in practice and certainly they are worthy of study in their own right. Though, note that Lucas et al. discuss beta VAEs in Chapter 5 and draw a clear connection between the role of beta and of the observation noise in the general ELBO objective with Gaussian likelihood. In particular, that when treating both as fixed hyperparameters the exact same gradient update can be computed in either case.
> - We now added the case when the decoder variance is learned and when beta is included. This makes the result of Lucas a strict subset of our results because their result is only applicable when (1) $\beta=1$, (2) $y=x$, and (3) $d_1<d_2$, whereas our updated analysis removes all three constraints. As you argued, it is important to consider the case when $\eta_{\rm dec}$ is learned and $\beta$ is a hyperparameter, and our results are thus more relevant and general than the previous works.
>
> > The remaining contribution of this work that is not supported by prior work is that the authors consider more general models than VAEs (with different output dimensions). However, I do not consider this contribution to be significant enough to overcome the negative points discussed elsewhere in my review.
> - With the updated results, we are confident that this criticism no longer applies.
>
> > Missing discussion of related work
> - Thanks for pointing out the relevant works. We have now added all five works to the reference and discussed them where appropriate.
>
> > I feel that the following is too strong a statement: "Up to now, there has not been any precise identification of neither the nature nor the cause of the posterior collapse problem." See discussion above.
> - We softened this statement to “Up to now, the study of nature or the cause of the posterior collapse problem is limited.”
>
> > A is positive semi-definite but it is then assumed that A1/2  is invertible. This is equivalent to assuming that all eigenvalues of the data covariance matrix are strictly positive, and it is stated in footnote that it is straightforward to extend to the low-rank case. But this is not evident to me from the proofs. For such a claim, ideally, the proofs in the supplementary material would apply to the low-rank case. You could keep the assumption in the main paper if it heavily simplifies notation etc., but I do not quite see how.
> - We have now updated Proposition 1 and the related parts to include the case when $A$ is not full-rank.
>
> > Why does notation appear at the end of the problem setting section? There is some duplication here, as much of it has already been defined above.
> - The point is just to summarize the notation in case the readers need a quick reference.
>
> >There is some confusing notation in the statement of Proposition 1. It swaps between U and W with U∗ doubly defined as the optimum of both objectives implicitly. Notation is also swapped in the proof. Further, the proof of Proposition 1 claims that these minimization problems are identical. It is have shown that, L_VAE(U,W)=L(U,A1/2W), which is not quite the claim (particularly in the case where A is not full-rank, then each minimizer of the former may correspond to a manifold of solutions for the latter). The proposition statement compares the minimizers, but note that both objectives are non-convex and a unique minimizer has not yet been established. Perhaps a more accurate statement is that any stationary point has these dual forms (under a full-rank A).
> - We have updated Proposition 1 and its proof to make it applicable to the case when A is not full-rank.
>
> > Proof of proposition 2. The proof begins by stating conditions on the global minimum; but this is a non-convex optimization problem and so this condition guarantees only a stationary point. As written, this is not incorrect but is a little confusing. The conditions are established for a stationary point and then solved for the global minimum by finding the stationary parameters with the smallest loss.
> - We updated the writing in the proof of proposition 2
>
> > I encourage the authors to state the SVD of Z within the proof again (otherwise the definition of F can be a source of temporary confusion).
> - We added the supplement information about SVD of Z in the proof of proposition 2

---

> ### Author Response · Authors · 2022-08-02
> **Author Reply Part 3**
>
> > Please state the trace inequality that you are using exactly (trace of matrix product upper bounded by sum of product of eigenvalues). It is also worth including the derivation of equation 45 --- the proof is difficult to follow without this. Similarly for equation 46 (how did you verify that only the larger solution is related to the minimum?).
> - We added a description of the von Neumann inequality and how this equality is applied in the proof of proposition 2
>
> > Proof of proposition 3. It is written that the gradient is an increasing function and so the minimum value is determined when the gradient is zero. You also need to add the condition that ℓ′(σ)<0 for some σ which is then given in closed-form.
>
> - We added closed-form $\sigma_+$ and $\sigma_-$ such that $l’(\sigma_+) > 0$ and $l’(\sigma_-) < 0$. Therefore $l’(sigma) = 0$ indicates the global minimum.
>
> Minor comments:
>
> > L23: in what sense is this more general? Due to x and y? Or due to the stochastic reparameterization trick used in VAEs? There is also no marginalization over y and it is unclear what x and y represent here. (This is corrected later, in Section 3).
> - We have now modified this related sentence and fixed the expression.
>
> > L34: "in the deep learning context" -> suggests you are doing this. But it is a stretch to claim this when studying linear models only.
> - We modified this sentence to avoid overclaiming.
>
> > L44-45: a strong claim given referenced work. May need further justification.
> - Our updated analysis of the case when $\eta_{\rm dec}$ is learnable suggests that the results of Lucas et al. are insufficient to identify the cause of the collapse, and this claim is, in fact, strengthened by the additional analysis.
>
> > L245: your posterior collapse condition also depends on the decoder variance. You also write above that the decoder variance can also be controlled to alleviate posterior collapse. And finally, you do not analyze learning the decoder variance but if you included the partition function in the objective I expect that the same analysis of Lucas et al. could be repeated in this setting to the same general effect.
> - See the newly added analysis of the case of a learnable $\eta_{dec}$ in Appendix Section C. Our result suggests that being able to tune $\beta$ can still be very important.
>
> > L284: Similar to L245, this is only true when excluding the partition function from the objective (and restricting what we mean by "the effect").
> - True. We have now updated this statement according to the newly added theoretical analyses.
>
> > L326-327: "Moreover, can we design a Bayesian-principled method for avoiding posterior collapse that does not involve hand-tuning β or ηdec?" This is exactly one of the contributions of Lucas et al., in the case of the linear VAE.
> - As we have argued above and in the newly added theoretical analysis, the suggestion of Lucas et al. is insufficient to fully remedy the problem of posterior collapse.

---

> > ### Comment · Reviewer_R7JS · 2022-08-09
> > **Response to rebuttal**
> >
> > Thank you for your detailed response. I also apologize for the late reply --- there were significant changes to the paper and a lot of details to carefully review. I'd also like to apologize that some of my criticisms came off as harsh. This was certainly not my intention.
> >
> > I appreciate the authors' various clarifications and the new additions to the paper regarding related work and new theoretical contributions. In particular, tackling learnable encoder variance and low-rank design matrices are valuable novel contributions. For brevity, I limit my response to only the most relevant issues.
> >
> > I did not review Appendix C in detail. Though, from my quick look-through, this appears to be a valuable contribution that pushes beyond existing prior work.
> >
> > > First of all, we point out that Eqs. (2)-(5) are correct. They are just different from those of Lucas et al..
> >
> > My issue was both technical and philosophical. Technically, Equation (3) contains the expected value of $\log(p(y∣z))$, by definition this term, corresponding to the log density of a Gaussian, includes a partition function. Equality is used in the further derivation of the objective but the partition function is dropped. The philosophical issues that I raised in my review will be discussed further below.
> >
> > ### Appendix D
> >
> > I must admit that I had a hard time following along with Appendix D. The proof is long and detail-oriented, perhaps necessarily. Early on, you take the optimal form of the parameters given in Proposition 1 to form the objective $G(s)$ (Equation 44). But the objectives no longer exactly align when we allow $\eta_{\textrm{dec}}$ (now $s$) to be a learnable parameter (as you have removed the $1/s$ factor in front). Nonetheless, I believe that the derivation is correct with these issues falling away in the details.
> >
> > I agree that this analysis is interesting and am more convinced that the inclusion of $\beta$ changes the optimal solution non-trivially. I would like to point out that the general finding that $\beta=1$ produces posterior collapse in the case where the $d - k + 1$ minor eigenvalues are equal is already known; see Appendix A.4 in the pPCA paper. In this case, posterior collapse _is_ the maximum likelihood solution. Further, this case is generally regarded as uninteresting (as stated in the pPCA paper) and, I would assume, is never the setting for posterior collapse occurring in deep latent variable models practically.
> >
> > Overall, I would still push back on the idea that "being able to tune $\beta$ can still be very important" for **linear VAEs**. However, speculatively, these results do perhaps shine some light on why tuning $\beta$ remains important for deep VAEs.
> >
> >
> > ### Remaining comments
> >
> >
> > > First of all, the suggestion that “there is no need for a beta parameter as all low-rank stationary points become saddle points” is incorrect.
> >
> > I now agree with the authors on this point.
> >
> > > if applying A (making decoder variance learnable) can remedy B (posterior collapse), then [not] A causes B.
> >
> > I am worried that I am missing your point here. At no point did I imply that fixed decoder variance is the cause of posterior collapse. My point was that if the problem is remedied, then $\beta$ becomes unimportant. The new Appendix D investigates this issue.
> >
> > The conclusion of Appendix D, as it pertains to this point, is that there _is_ a case where the linear VAE (and pPCA) have posterior collapse when $\beta=1$. I would argue that this case is relatively uninteresting; in the sense that users of linear VAEs need not worry about it for any realistic data sets. However, the apparent difference when including $\beta$ certainly makes it worthy of study and I appreciate the authors going through the effort of proving this in response to my claim.
> >
> > For what it is worth, I would be more precise in the main paper. The following sentence in your rebuttal; "The global minimum solution becomes low-rank when the smallest $k+1$ eigenvalues of $\Lambda$ are identical", is considerably more informative than the current sentence in the main paper: "Our analysis shows that even if $\eta_{\textrm{dec}}$ is learnable, posterior collapse can happen for some datasets." I'd suggest replacing "for some datasets" with "when the smallest $k+1$ eigenvalues of $\Lambda$ are identical" (and also cite Tipping & Bishop).
> >
> > ### Summary
> >
> > The paper has been significantly improved by the authors. The author's addressed all of my biggest concerns and I have decided to increase my score to reflect this. I still feel that the presentation of the paper could be improved; ensuring that all of the mathematical details are correct, providing some more detail in proofs, and .
> > Minor comment:
> >
> > - The form of the data-dependent variance in Appendix C is non-standard in practice (more typical would be to write $\sigma(x) = e^{Cx + f}$).
> > - L538 in Appendix D (and elsewhere), the partition function is missing factors of pi.

---

> > > ### Author Response · Authors · 2022-08-09
> > > **Reply**
> > >
> > > Thanks for the detailed feedback. We will improve our manuscript further in the final version.

---

### Official Review · Reviewer_gw2z · 2022-07-13

**Rating:** 8
**Confidence:** 3
**Soundness:** 4 excellent
**Presentation:** 4 excellent
**Contribution:** 4 excellent

**Summary:**

The work identifies the cause and the precise condition of posterior collapse of a linear latent variable model, and proposed methods to alleviate the problem by making the latent variance learnable.

**Questions:**

Some minor problems with the writing and derivations, for which the reviewer hopes the author(s) could clarify or fix:

Typos:
1. Line 117, $\Psi$ should be $\Sigma_Z$.
2. Line 140 and Eq (7), $U=F\Lambda P$, but in Line 466 of appendix, $U=Q\Lambda P$. Also F has been used by SVD of Z in Line 116 and in proofs. Please fix U's definition here to make things consistent.
3. Line 151 and 152, $\sigma_i$ should be $\zeta_i$, as introduced in Line 118.
4. Line 200, it should be $\mathrm{min}(d_0, d_2)$ instead of $\mathrm{min}(d_1, d_2)$

In Appendix:

 5. Line 444, the mean should be $U z + b_d$.
 7. Eq 24, should be $\frac{\partial L_{VAE}}{\partial b_d}$.
 6. Eq 36, should be $VU^TU - Z^TU + \cdots, sign is wrong for the second term.
 7. Line 469, $\Sigma$ should be $\Sigma_Z$. Line 470, $\sigma_i$ should be $\zeta_i$ as of the singular value $\Sigma_Z$.
 8. Eq 52, $\zeta^2_i$ should be $\zeta_i$.

Derivations:

 9. For proof of Corollary 1, Appendix Eq (49) is correct. But going from App. Eq(49) to Eq(11) is unclear to the reviewer.
 10. The proof of Proposition 4, in appendix. To reach Eq (23), it already used $b^*_{d}=E_x[y]$ The correct proof should be put together two equations for $\frac{\partial L_{VAE}}{\partial b_d}=0$ and $\frac{\partial L_{VAE}}{\partial b_e}=0$, two equations together to reach the $b^*_{d,e}$ solutions.
 11. For Eq (44), the author(s) should add the condition, when the equality holds for von Neumann's Trace Inequalities. (btw, it's "von Neumann, "v" without capitalisation)

Some of questions to help the reviewer understand better:

1. To the understanding of the reviewer, he believes that there is a physical interpretation of the term $\frac{\sqrt{\beta}\sigma_i\eta_{dec}}{\eta_{enc}}$. Inspired by the author(s) to explain "$\zeta_i$ as the strength of the alignment between the input $x$ and the target $y$" (Line 152), the term could be explained as the relative variance of the alignment between the model prediction. Then the thresholding in Eq(8) is to encourage U and V to be low-ranking by discarding the modes that the model variance scale $\frac{\sqrt{\beta}\sigma_i\eta_{dec}}{\eta_{enc}}$ is higher than the data signal $\zeta_i$. Is the understanding correct?

2. In Line 169, "we have a complete posterior collapse: both U and V are identically zero". It seems suffice to have posterior collapse with V (henceforth W as in line 97) being zero. Is there any significance for U being identically zero?

3. In line 176, "one alternative way to fix posterior collapses ... is to use a sufficiently small $\eta_{dec}$". Does it mean a more deterministic decoder is less likely to have posterior collapse? If so, does it generalise to nonlinear models?

**Limitations:**

The author(s) discussed some limitations and future work plans in 6. Outlook.

**Strengths And Weaknesses:**

 * Originality: The theoretical analysis and the direct pinpointing of the cause of posterior collapse problem for linear latent variable models are novel and insightful.
 * Quality: The derivation and theoretical analysis are sound.
 * Clarity: The paper was written in a clear and concise way, with detailed derivation to prove the propositions/corollary/theorems in the appendix. Some minor fixes are required, for which the reviewer listed in the Questions section.
 * Significance: Better understanding of the posterior collapse is important not only for theoretical understanding of deep learning, but also for designing better principles to build models and training strategies to improve training stabilities. This work is important on this front.
Overall, the paper is theoretically sound, well written and carefully explained. The reviewer enjoys the reading.

---

> ### Author Response · Authors · 2022-08-02
> **Author Reply**
>
> Thank you very much for pointing out the typos and unclear points. We have fixed the typos in the updated manuscript. The following are the answers to your specific questions.
>
> > To the understanding of the reviewer, he believes that there is a physical interpretation of the term $\frac{\beta\sigma_i \eta_{dec}}{\eta_{enc}}$. Inspired by the author(s) to explain $\zeta_i$ as the strength of the alignment between the input $x$ and the target $y$ (Line 152), the term could be explained as the relative variance of the alignment between the model prediction. Then the thresholding in Eq(8) is to encourage U and V to be low-ranking by discarding the modes that the model variance scale $\frac{\beta\sigma_i \eta_{dec}}{\eta_{enc}}$ is higher than the data signal $\zeta_i$. Is the understanding correct?
> - Yes. You are correct to interpret $\frac{\beta\sigma_i \eta_{dec}}{\eta_{enc}}$ as the model variance scale. However, we would not call it “relative” because we have shown that $\sigma_i$ roughly scales as $\eta_{enc}$, and so the model is more like the “absolute model variance scale.”
>
> > In Line 169, "we have a complete posterior collapse: both U and V are identically zero". It seems suffice to have posterior collapse with V (henceforth W as in line 97) being zero. Is there any significance for U being identically zero?
>
> - It is true that $W=0$ is sufficient to cause the complete collapse. Here, we mention $U$ and $W$ together because they are always zero simultaneously. $U=0$ has the effect of reducing the reconstruction variance to zero and is thus encouraged when $W=0$.
>
> > In line 176, "one alternative way to fix posterior collapses ... is to use a sufficiently small $\eta_{dec}$." Does it mean a more deterministic decoder is less likely to have posterior collapse? If so, does it generalise to nonlinear models?
> - Thanks for asking this question. It is not correct to say that a deterministic decoder is less likely to collapse. For example, see https://arxiv.org/abs/1901.08168 for how a deterministic autoencoder could have a collapse-like behavior when $L_2$ regularization. Mathematically speaking, the case of a VAE is similar to that of an $L_2$-regularized autoencoder because the randomness in the model causes an effective $L_2$-regularization. Thus, collapses are not unique to stochastic models.

---

### Author Response · Authors · 2022-08-02
**Summary of revision**

We thank all the reviewers for their careful and constructive suggestions. In the update, we made significant updates to (1) make our theoretical analysis more comprehensive, (2) remove overclaiming, and (3) improve the discussion of the previous works.
Content-wise, we significantly extended the theoretical analyses in the appendix to include cases asked about by the reviewers, making our work as comprehensive as possible. The major additions are colored in orange. We have added the following analyses
1. when the decoder variance is learnable as in Lucas et al. (2019) [Appendix Section D]
2. when the data covariance $A$ is not invertible [Updated Proposition 1 and its proof]
3. when the encoder variance is dependent on the input, which is common in practice [Appendix Section C]

With these updates, we are confident that our work presents a solid and significant theoretical contribution to the community. In our specific replies to the reviewers, we clarify the misunderstandings or the points that were unclear. Also, if the reviewers point out any remaining problems, we are willing to make revisions to the manuscript accordingly.

---

### Meta-Review · Area_Chair_kejR · 2022-08-25

**Recommendation:** Accept
**Confidence:** Certain

**Metareview:**

This paper analyzes the phenomenon of posterior collapse in linear variational autoencoders.  While only the linear case is addressed, all reviewers found the work worthy of acceptance, citing its clear contributions to this line of literature that seeks to understand how deep architectures interact with the evidence lower bound.  In particular, this paper (for the linear model class) is able to pinpoint collapse to regularization of the mean of the latent variables.

**Award:**

No

---

### Decision · Program_Chairs · 2022-09-14

Accept